behaviour, biomechanics, ecology

tissue body density, UAV, feeding season, animal-borne sensor, neutral buoyancy, cetacean

**Authors for correspondence:**
Kagari Aoki
e-mail: aokikagari@aori.u-tokyo.ac.jp
Patrick J. O. Miller
e-mail: pm29@st-andrews.ac.uk

# Aerial photogrammetry and tag-derived tissue density reveal patterns of lipid-store body condition of humpback whales on their feeding grounds

Kagari Aoki[1], Saana Isojunno[2], Charlotte Bellot[3], Takashi Iwata[1], Joanna Kershaw[2], Yu Akiyama[1], Lucía M. Martín López[2,4], Christian Ramp[2,5], Martin Biuw[6], René Swift[2], Paul J. Wensveen[2,7], Patrick Pomeroy[2], Tomoko Narazaki[1], Ailsa Hall[2], Katsufumi Sato[1] and Patrick J. O. Miller[2]

[1]Atmosphere and Ocean Research Institute, The University of Tokyo, Kashiwa, Chiba 2778564, Japan
[2]Sea Mammal Research Unit, School of Biology, University of St Andrews, St Andrews, Fife KY16 8LB, UK
[3]Department of Marine Biology, University of Neuchâtel, Neuchâtel 2000, Switzerland
[4]Asociación Ipar Perspective, Sopela 48600, Spain
[5]Mingan Island Cetacean Study (MICS), St. Lambert, Quebec, Canada G0G 1V0
[6]Fram Centre, Institute of Marine Research, Tromsø N-9296, Norway
[7]Faculty of Life and Environmental Sciences, University of Iceland, 102 Reykjavik, Iceland

 KA, 0000-0002-5986-9849; SI, 0000-0002-2212-2135; PJW, 0000-0002-9894-2543; PJOM, 0000-0001-8028-9673

Monitoring the body condition of free-ranging marine mammals at different life-history stages is essential to understand their ecology as they must accumulate sufficient energy reserves for survival and reproduction. However, assessing body condition in free-ranging marine mammals is challenging. We cross-validated two independent approaches to estimate the body condition of humpback whales (*Megaptera novaeangliae*) at two feeding grounds in Canada and Norway: animal-borne tags ($n = 59$) and aerial photogrammetry ($n = 55$). Whales that had a large length-standardized projected area in overhead images (i.e. whales looked fatter) had lower estimated tissue body density (TBD) (greater lipid stores) from tag data. Linking both measurements in a Bayesian hierarchical model to estimate the true underlying (hidden) tissue body density (uTBD), we found uTBD was lower ($-3.5 \text{ kg m}^{-3}$) in pregnant females compared to adult males and resting females, while in lactating females it was higher ($+6.0 \text{ kg m}^{-3}$). Whales were more negatively buoyant ($+5.0 \text{ kg m}^{-3}$) in Norway than Canada during the early feeding season, possibly owing to a longer migration from breeding areas. While uTBD decreased over the feeding season across life-history traits, whale tissues remained negatively buoyant ($1035.3 \pm 3.8 \text{ kg m}^{-3}$) in the late feeding season. This study adds confidence to the effectiveness of these independent methods to estimate the body condition of free-ranging whales.

## 1. Introduction

Accumulating sufficient energy from the environment affects both survival and breeding success for many animal species, and thereby influences the dynamics of entire populations [1]. Required energy stores can vary with season, sex, age class, reproductive stage and food availability [2–6]. Mammalian body condition improves with increased lipid stores in fat or blubber [7] often represented as the ratio between body lipid and lean tissue mass [1,8]. Individuals in good body condition with larger energy stores generally have better

resilience to environmental variability, and higher survival rates than individuals in poor condition [9]. The pregnancy rate or reproductive success of mammals declines when energy levels are insufficient [3,5,10]. Therefore, body condition provides a key dynamic state variable with direct consequences for reproductive output, fitness and demography of free-ranging mammals, with the potential to assess how human activity and environmental changes impact individuals and populations [11,12].

Body condition can be quantified using morphological measurements (e.g. mass/length ratios), biochemical analysis of organs or body fluids (e.g. blood composition) and physiological condition (e.g. blubber thickness) [1,13]. For mammals that can be temporarily captured, isotope dilution is a preferred technique to quantify total lipid content [14]. An alternative non-invasive method is to measure the physical body density (i.e. mass/volume). The lipids of mammals are less dense (920 kg m$^{-3}$) than skeletal muscle (ca 1060 kg m$^{-3}$), proteins (140 kg m$^{-3}$) or water (1000 kg m$^{-3}$) [15,16]. Because lipid stores are less dense, they require more volume than proteins and muscle per unit mass, therefore, mammals with a high percentage of lipids have lower overall tissue body densities ($\rho_{tissue}$), while those with greater protein stores have higher body densities. Changes in total body density can thus reflect changes in total lipid mass, which may not always be reflected in morphometric body condition indices [13].

Cetaceans and pinnipeds store most of their energy reserves as lipid in blubber, which may represent up to 50% of their body mass in certain life stages [4]. Blubber is an important adaptation for aquatic life: it functions as a thermal insulator, contributes to water balance, streamlines the body and serves as an elastic spring for efficient locomotion [17]. Marine mammals with large proportions of lipid have lower $\rho_{tissue}$ and are more buoyant [18–20]. Gases in the body also increase positive buoyancy, particularly at shallow depths where they are less compressed [21].

Buoyancy influences the stroking patterns of diving mammals, with gliding behaviour increasing when net buoyancy aids movement [18–22]. Fatter, more buoyant seals predominantly perform stroke-and-glide swimming during both descent and ascent, whereas leaner and negatively buoyant seals perform prolonged glides during descent and stroke continuously during ascent [18]. Thus, buoyancy influences round-trip locomotion cost to and from depth, with neutral buoyancy thought to be the most efficient for minimizing round-trip costs [19,23]. Therefore, in leaner negatively buoyant diving mammals, accumulation of low-density lipids shifts the body towards neutral buoyancy, reducing cost-of-transport. In fatter positively buoyant individuals, a further increase in low-density lipid stores may increase cost-of-transport, leading to trade-off between energy reserve accumulation and locomotion costs.

Baleen whales are the largest predators on earth, and some species undergo distinct seasonal migrations between high-latitude summer feeding grounds and low-latitude winter breeding grounds [24–26]. Most migratory baleen whales are 'capital-breeders', which must accumulate large amounts of energy at the feeding grounds for fasting during migrations and breeding. The quantity of energy stored as lipid is a good predictor of survival, pregnancy rates [1] and, offspring body condition, growth and survival [27,28].

Traditional approaches to examine variation in the energy stores of baleen whales involved collecting anatomical measurements, often obtained during whaling operations [4,29,30]. Recent advances in high-resolution tag data have allowed researchers to estimate tissue body density (TBD) of various free-ranging diving animals by analysing hydrodynamics [20–22,30–33] to estimate drag and buoyancy forces acting on the animal body during descent and ascent glides. This 'glide method' has been applied widely to tagged sperm whales (*Physeter macrocephalus*) [21], northern elephant seals (*Mirounga angustirostris*) [31], northern bottlenose whales (*Hyperoodon ampullatus*) [32], long-finned pilot whales (*Globicephala melas*) [20] and humpback whales (*Megaptera novaeangliae*) [33]. Aoki *et al.* [31] validated the glide method against isotope dilution [14,34] of northern elephant seals. TBD estimated using the glide method matched that expected from fat content measured by isotope dilution and experimental manipulation of body density using weights and floats. Large cetaceans cannot be captured for isotope-dilution measurements, so alternative methods of estimating body condition have been developed [35]. Blubber thickness in the North Atlantic right whale (*Eubalaena glacialis*) varied with reproductive status [36]. Qualitative visual assessments of body condition from photos can provide viable proxies for certain species [37]. While most baleen whale lipid reserves are stored in blubber, considerable amounts are also stored in muscle and intra-abdominal fat [4,38]. A unique advantage of the TBD method is that it captures the buoyancy effect of total body lipid stores simultaneously.

Aerial photogrammetry has proved successful for measuring changes in cetacean body shape in relation to reproductive status [27,28,39,40]. Unmanned aerial vehicles (UAVs) are particularly useful for monitoring wildlife and habitats in places that are difficult to access or navigate from the ground, as well as approaching sensitive or aggressive species [41,42]. Increased lipid stores in blubber increase the girth of cetaceans, particularly for pregnant females which require large lipid stores to support lactation [40]. Christiansen *et al.* [28] obtained repeated measurements of several southern right whales (*Eubalaena australis*) and used UAV images to recognize animals using their distinctive natural markings. Width measurements were used to model changes in body volume of females during the breeding season. The energetic cost of lactation was apparent from females' decreased body volume, which contrasted with increasing length and volume of their calves, confirming that maternal investment towards the growth of calves of large free-ranging cetaceans could be measured [28]. Longer, fatter mothers expended more and produced larger calves, demonstrating the importance of lipid accumulation during the feeding season for successful breeding.

Here, we investigate lipid-store body condition patterns in humpback whales in two geographically distinct feeding grounds in the Atlantic Ocean (figure 1): eastern Canada (northwest Atlantic) and Norway (northeast Atlantic). Most humpbacks in both locations probably breed in the West Indies during winter [43]. Consequently, whales that forage in Norway annually migrate 2–3 times further one-way trip distance (approx. 8500–9500 km) than whales that forage in Canada. Longer duration and migration distance have a greater energetic cost; therefore, we expected that Norwegian whales might have greater lipid stores in the late feeding season, but lower stores at the early feeding season. A key objective was to cross-validate two different non-invasive methods to measure body condition: TBD estimation using

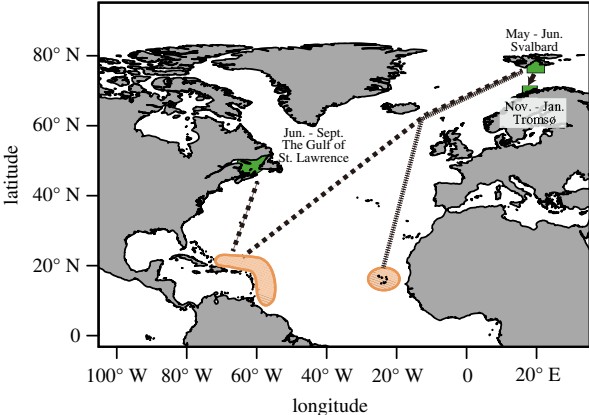

**Figure 1.** Study locations (green) on two separate feeding grounds: Canada and Norway, with their breeding locations (orange diagonal lines: West Indies and possibly Cape Verde Islands). The field seasons are shown. The figure was modified from the map on the NOAA website (https://www.fisheries.noaa.gov/species/humpback-whale) and the NAMMCO website (https://nammco.no/topics/humpback-whale/). (Online version in colour.)

animal-borne tags and shape measurement from overhead photogrammetry images. We then assess an index of lipid-store body condition derived from these two methods varied in relation to sex, reproductive status, location and timing within the feeding season.

## 2. Material and methods

### (a) Study area and data collection

Data were collected from two geographically distinct humpback whale feeding grounds (figure 1; see electronic supplementary material). In the Gulf of St Lawrence, Canada, fieldwork was conducted from June to September in 2011–2012 [33] and 2016–2017. In Norway, fieldwork was performed off Svalbard in May to June, 2011–2012, and the Kaldfjorden basin (outside Tromsø) from November to January in 2013–2014 and 2016–2018. Non-invasive multi-sensor suction-cup tags were attached to individuals using either a 5 m hand-pole or a pneumatic launching system. In 2016–2018, we flew a UAV (DJI Phantom 4, DJI Co., Ltd., ShenZhen, China) to obtain overhead images of individual tagged whales and surrounding animals within the same frame. We attempted to obtain a biopsy sample of each tagged whale. A conductivity temperature depth (CTD) profiler (miniCTD; Valeport Ltd., UK) was used to measure seawater density near tagged animals, weather permitting, within 24 h of tag deployment and within 1 km of the deployment location. Life-history traits (age class, sex, reproductive status) were assessed from field observations, photo-identification, and genetic and hormone analyses [44] of biopsy samples collected from tagged whales.

### (b) Time-series data analyses

We extracted the following time-series variables from animal-borne recorders housed within the suction-cup attached tags (Little Leonardo loggers [31–33] or Dtags [45], electronic supplementary material): (i) depth derived from pressure, (ii) body orientation (pitch and roll) calculated from lower frequency acceleration, (iii) fluke strokes (i.e. dorsoventral oscillations) from higher-frequency acceleration, (iv) swim speed, measured using an external propeller on the Little Leonardo loggers, or by the rate of change in depth, divided by the Dtag-measured sine of pitch.

### (c) Hydrodynamic performance model

Based on Miller *et al.* [32], acceleration (m s$^{-2}$) along the swimming path of a gliding body is determined by drag (the first additive term), buoyancy force derived from the density of the non-gas component of the whale body (the second term) and buoyancy caused by residual air inside the animal (the third term):

$$\text{acceleration} = -0.5 \cdot \frac{C_D \cdot A}{m} \cdot \rho_{sw} \cdot v^2 + \left( \frac{\rho_{sw}}{\rho_{tissue}(d)} - 1 \right) \cdot g$$
$$\cdot \sin(p) + \frac{V_{air}}{m} \cdot g \cdot \sin(p)$$
$$\cdot \frac{\rho_{sw} - \rho_{air} \cdot (1 + 0.1 \cdot d)}{(1 + 0.1 \cdot d)},$$

where

$$\rho_{tissue}(d) = \frac{\rho_{tissue}(0)}{1 - r \cdot (1 + 0.1 \cdot d) \cdot 101325 \cdot 10^{-9}}. \quad (2.1)$$

Here, $C_D$ is the drag coefficient, $A$ is the relevant surface area (m$^2$), $m$ is the mass of the whale (kg), $\rho_{sw}$ is the density of the surrounding seawater (kg m$^{-3}$), $v$ is the swim speed (m s$^{-1}$), $\rho_{tissue}$ is the density of the non-gas component of the whale body (kg m$^{-3}$), $g$ is the acceleration due to gravity (9.8 m s$^{-2}$), $p$ is the animal pitch (radians), $V_{air}$ is the volume of air at the surface (m$^3$), $\rho_{air}$ is the density of air (kg m$^{-3}$), $d$ is the glide depth (m) and $r$ is the compressibility for animal tissue (i.e. the fractional change in volume per unit increase in pressure). Compressibility was fixed as $0.38 \times 10^{-9}$ Pa$^{-1}$ based on the value estimated for northern bottlenose whales [32]. The equivalent compressibility value for 0°C water of salinity 35 ppm is $0.45 \times 10^{-9}$ Pa$^{-1}$. The value 101 325 converts pressure in atmospheres to pressure in Pascals, so that the units of body tissue compressibility are proportion per Pa × $10^{-9}$.

### (d) Data processing and Bayesian estimation for the hydrodynamic model

We extracted the following variables during each 5 s glide from processed time-series tag data and a CTD cast: acceleration measured using linear regression of speed versus time, average pitch ($p$), swim speed ($v$) and seawater density ($\rho_{sw}$) (electronic supplementary material). The unknown parameters in the hydrodynamic glide model ($\rho_{tissue}$, $V_{air}m^{-1}$ and $C_DAm^{-1}$) were estimated by Bayesian Gibbs sampling with the freely available software JAGS within R (coda, R package v.0.17-1 2015, http://cran.r.project.org/web/packages/coda/index.html) and R2jags (R package v.0.5-7 2012, https://cran.r-project.org/web/packages/R2jags/index.html) using data extracted for each 5 s glide [32]. We report the mean and 95% percentile, termed posterior mean and 95% credible interval (CI), of the posterior samples as the best estimates of the parameter value and its uncertainty. Following Miller *et al.* [32], we compared models where parameter values were either specific to each individual ('individual' parameters) or shared across all individuals ('global' parameters). In addition, $V_{air}m^{-1}$ was allowed to vary between dives.

### (e) Estimating length-standardized surface area index from aerial photogrammetry images

We extracted multiple video frames per individual during surfacing and allocated a score (i—poor, ii—medium or iii—good) to each of three criteria: (i) animal posture, (ii) brightness and (iii) animal depth relative to the water surface (electronic supplementary material, table S1), and the photo with the highest average score was used. We measured whale length in pixels between the position of the tip of the rostrum and the

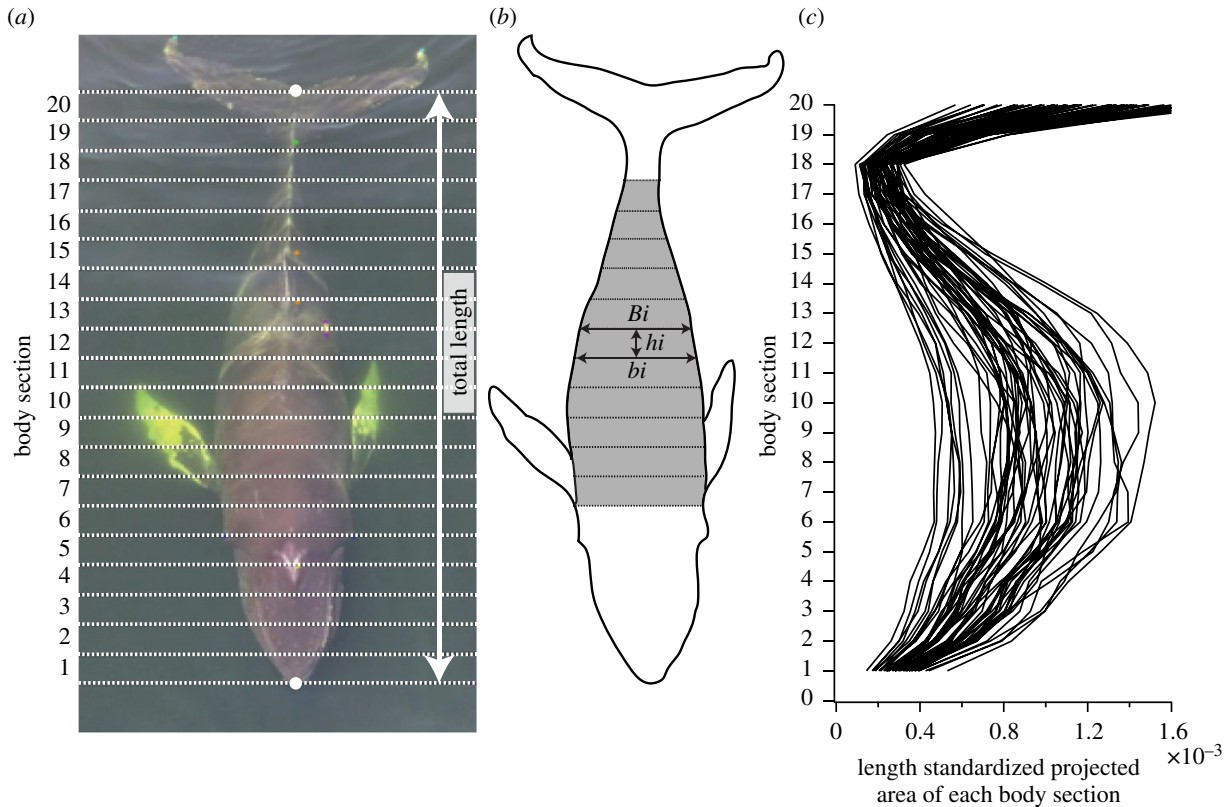

**Figure 2.** Length-standardized surface area index (LSSAI) calculated from aerial photogrammetry (see Material and methods). (a) The total length between the tip of the rostrum and fluke notch is divided into 20 equal length sections. (b) Grey shadows (body sections 7–17) indicate LSSAI. (c) Variation in each body section across whales. (Online version in colour.)

notch of the flukes and divided the length into 20 equal sections ([27], figure 2a). The external boundary of the whale was marked to measure the width of the whale (in pixels) at each section.

We expected animals with larger lipid stores to be visibly wider for a given body length than animals with smaller lipid stores [18]. To quantify this expected effect, we used the outline measurements of each whale to calculate length-standardized surface area index (LSSAI) based on Christiansen *et al.* [27]. The length-standardized projected area (PA) of each section of an individual whale's body was calculated using the trigonometric equation for the area of a trapezoid (figure 2b) as

$$PA = (B/L + b/L) \cdot \left(\frac{h}{L}\right)/2, \tag{2.2}$$

where $B$ and $b$ are the widths of the top and bottom of each section of the whale's body in pixels, respectively, and $h$ is the height of each region in pixels. To standardize the area by length, each dimension was divided by total length ($L$) in pixels. We summed PA over sections 7–17 (i.e. 35–85% body length from the rostrum) as preliminary data indicated these sections contained the most variation across whales (figure 2c, [27]):

$$LSSAI = \sum_{i=7}^{17} PAi \tag{2.3}$$

where $i$ is the number of each body section.

## (f) Estimates of underlying tissue body density: effect of date and reproductive status

The model specified a single process for an underlying, or 'hidden', TBD. This hidden process was a log-linear Gamma

regression model with the following covariates: location, Julian date, sex (factor covariate with three levels: male, female and unknown) and three presence/absence covariates describing reproductive status (lactating [females], pregnant [females] and immature). LSSAI and TBD estimates were treated as two different observations of underlying TBD (uTBD). The model estimated an intercept and slope that related LSSAI to TBD. Observation error was estimated in the model for LSSAI, while the observation error for body density was specified for each glide segment based on the hydrodynamic model error. Thus, this model avoided multiple testing and accounted for measurement error inherent to both metrics. Another advantage is that all data obtained using either approach could be used to estimate the effect of the covariates on body condition as the model predicted any missing observations of TBD or LSSAI based upon the specified relationships with uTBD. See the electronic supplementary material R script for further details.

## 3. Results

We recorded the fine-scale underwater movement of 70 humpback whales, with 732 h dive data. We excluded 11 whales with less than 10 glides, leaving 59 whales with sufficient glides for estimating TBD (electronic supplementary material, table S2). Thirty-two whales were from Canada and 27 whales were from Norway.

Eleven hours of UAV footage (averaging 13 min individual$^{-1}$) were obtained for 55 individuals to calculate LSSAI from overhead images: 21 tagged animals (table 1) and 34 non-tagged animals (electronic supplementary material, table S3) that were in the same frame as a tagged whale.

## (a) Tissue body density estimates from using hydrodynamic analysis

The individual-average (global) tissue density of the most parsimonious Bayesian model with the lowest deviance information criterion (electronic supplementary materials) was estimated as $1037.2 \pm 3.0$ (mean ± CI) kg m$^{-3}$. Individual posterior mean values for tissue density varied across individuals in both feeding areas (range: $1029.1$–$1049.6$ kg m$^{-3}$ and $1027.9$–$1050.9$ kg m$^{-3}$ in Norway and Canada, respectively). The TBD of all animals in both feeding areas was greater than that of the corresponding seawater density ($1027.3 \pm 0.7$ kg m$^{-3}$ in Norway; $1023.3 \pm 2.0$ kg m$^{-3}$ in Canada), indicating that non-gas body tissues were denser than seawater.

The posterior mean of the global drag term ($C_D\,Am^{-1}$) was $12.7 \pm 3.6 \times 10^{-6}$ m$^2$ kg$^{-1}$, overlapping with the mean of the specified normal prior ($11 \times 10^{-6}$ m$^2$ kg$^{-1}$), consistent with drag being partly induced by lift [33]. Posterior means for the drag term for most individuals were $5$–$25 \times 10^{-6}$ m$^2$ kg$^{-1}$ (range: $0.3$–$37.0 \times 10^{-6}$ m$^2$ kg$^{-1}$; electronic supplementary material, table S2). Relatively large flippers and shallow pitch angles during descent and ascent (absolute value, $49.1 \pm 13.7°$, $n = 6602$ glides) probably caused greater induced drag. The global mean air volume was estimated as $37.3 \pm 1.6$ ml kg$^{-1}$.

## (b) Repeated measurements of tissue body density for three individuals

The TBD of three individuals was measured twice across seasons or years in Canada (ID H002, H584, H607 as also analysed here; electronic supplementary material, table S2, see Narazaki *et al.* [33] for adult male ID H607). These whales' body condition differed according to the time when measurements were obtained in the feeding season, and by reproductive status. Changes in gliding patterns of these two individuals (ID H584 and H002) corresponded with the TBD (figure 3 and electronic supplementary material, figure S1).

Female H002: TBD was higher ($1050.9$ kg m$^{-3}$, i.e. low lipid stores) when resting (not pregnant nor lactating) during the early part of the feeding season in 2016, compared to when she was pregnant in the mid feeding season of 2011 ($1027.9$ kg m$^{-3}$).

Female H584: TBD was relatively low ($1028.8$ kg m$^{-3}$) when pregnant during mid-feeding season in 2011, indicating a large lipid store. When lactating during the early part of the feeding season of 2017, her TBD was comparatively higher ($1035.7$ kg m$^{-3}$).

## (c) Underlying tissue body density in relation to date and breeding status

LSSAI and TBD were negatively correlated (Pearson correlation, $r = -0.48$, $p = 0.0265$) in 21 whales for which both indices were available, indicating that, as expected, animals with a greater projected surface area (LSSAI) had lower TBD (greater lipid stores, figure 4).

For the hierarchical model of seasonal changes in uTBD, we used all 93 whales for which either LSSAI, TBD or both indices were available (figure 5). uTBD did not differ substantially between adult females and males (females $1037.4 \pm 1.1$ kg m$^{-3}$; males $1037.2 \pm 1.2$ kg m$^{-3}$), but uTBD decreased throughout the feeding season ($-2.7$ kg m$^{-3}$ 100 d$^{-1}$), indicating an expected increase in lipid stores. The uTBD was lower for pregnant females ($-3.5 \pm 1.6$ kg m$^{-3}$), indicating greater lipid stores than adult resting females and males, while it was higher, indicating lower lipid stores, for lactating females ($+6.0 \pm 2.6$ kg m$^{-3}$) compared to adult females and males. Furthermore, uTBD was higher in Norway compared to Canada early in the feeding season ($+5.3 \pm 1.2$ kg m$^{-3}$, i.e. model intercept, figure 5). Late in the feeding season (greater than 190 Julian days) uTBD did not differ by location (Canada, $1035.6 \pm 1.8$ kg m$^{-3}$ ± s.d., $n = 21$ whales; Norway, $1035.8 \pm 2.2$ kg m$^{-3}$, $n = 27$). Overall, uTBD was consistently greater (range: $1027.9$–$1049.5$ kg m$^{-3}$ in Canada and $1029.7$–$1049.6$ kg m$^{-3}$ in Norway) than seawater density ($1023.3 \pm 2.0$ kg m$^{-3}$ in Canada and $1027.3 \pm 0.7$ kg m$^{-3}$ in Norway).

# 4. Discussion

We successfully cross-validated two independent metrics of lipid-store body condition in a free-ranging cetacean: (i) TBD estimated by fitting the hydrodynamic glide model with high-resolution tag data, and (ii) projected surface area-to-length ratios (LSSAI) estimated using UAVs. The correlation between these metrics demonstrated the validity of using the hydrodynamic glide model to determine individual and temporal variations in the TBD of humpback whales. This method is suitable for aquatic animals that glide, including the relatively shallow-diving humpback whales evaluated in our study. Conversely, the cross-validation confirmed supported the less invasive aerial photogrammetry method, which is widely used in ecology to provide measurements of animals that are difficult to access (e.g. [28,37,39–42,46]). Aerial photogrammetry is used for measurement of body length [42,46], body condition including body width [39], body surface area [37] and body volume [28]. While the external body shape of an animal is an indicator of body condition, there have been no studies that demonstrate how external body shape relates to TBD. The external body shape and lipid content of the outer blubber layer (sampled by biopsy) has a low correlation in cetaceans [44,47,48]. Our analysis confirmed that body shape is linked to buoyancy changes, explained by the proportion of total body lipid stores. While aerial photogrammetry is widely applicable, the glide model enables in-depth studies on buoyancy and behaviour of diving vertebrates.

Hydrodynamic glide models have been used to estimate the TBD of elephant seals [31], sperm whales [21], northern bottlenose whales [32] and long-finned pilot whales [20]. Narazaki *et al.* [33] applied this method to relatively shallow-diving humpback whales; the gliding patterns of whales correlated with their estimated TBD, with denser whales gliding more while descending and less dense whales gliding more while ascending. Although the correlation between TBD and gliding patterns indicates the consistency of TBD estimates [33], cross-validation with LSSAI using photogrammetry images shows that shape and buoyancy changes are linked. LSSAI and TBD correlated negatively (figure 4), indicating that animals with a greater projected area had a lower TBD (i.e. greater lipid stores). We conclude that the residual of 'projected surface area' of mature humpback whales during their feeding season is mostly derived from total lipids stores, and hence energy stored in the blubber, rather than protein-based tissues.

**Table 1.** Detailed information of 21 humpback whales used for a comparison between tissue body density ($\rho_{tissue}$) from tag data and length-standardized surface area index (LSSAI) from aerial photogrammetry. The combined drag term ($C_D A m^{-1}$) obtained from the Bayesian estimation is also presented. Data ID was named for Tag data and UAV data separately. See the electronic supplementary material, table S2 for details of all 59 individuals used to estimate TBD. See the electronic supplementary material, table S3 for details of all 55 individuals used to calculate LSSAI.

| tag ID | UAV ID | whale ID | date | location | age class | sex[a] | no. of 5 s glides | $\rho_{tissue}$ (kg m$^{-3}$) | $C_D A m^{-1}$ ($\times 10^{-6}$ m$^2$ kg$^{-1}$) | LSSAI |
|---|---|---|---|---|---|---|---|---|---|---|
| Mn16_175a | DAR | H140 | 23 Jun 2016 | Canada | adult | F | 88 | 1032.9 ± 1.6 | 10.1 ± 4.2 | 0.07180 |
| Mn16_178a | HAN | — | 29 Jun 2016 | Canada | juvenile | F | 10 | 1043.0 ± 8.4 | 17.5 ± 12.3 | 0.06629 |
| Mn16_250a | BOO | H494 | 6 Sep 2016 | Canada | adult | F | 28 | 1036.8 ± 4.7 | 15.7 ± 4.6 | 0.07924 |
| Mn16_258a | TRA | H109 | 14 Sep 2016 | Canada | adult | F | 49 | 1028.8 ± 3.2 | 16.1 ± 6.4 | 0.09396 |
| Mn17_022a | HW1 | — | 22 Jan 2017 | Norway | adult (pregnant) | F | 35 | 1029.1 ± 1.2 | 10.8 ± 3.5 | 0.08181 |
| Mn17_026LLa | HW7 | — | 26 Jan 2017 | Norway | adult (pregnant) | F | 14 | 1033.9 ± 3.5 | 4.3 ± 6.7 | 0.08713 |
| Mn17_026a | HW8 | — | 25 Jan 2017 | Norway | adult | U | 84 | 1035.4 ± 1.2 | 7.6 ± 2.5 | 0.07129 |
| Mn17_158a | FOF | — | 6 Jun 2017 | Canada | adult | F | 83 | 1030.6 ± 1.8 | 8.2 ± 4.5 | 0.08463 |
| Mn17_165a | BOL | H102 | 12 Jun 2017 | Canada | adult (pregnant) | F | 131 | 1032.5 ± 2.2 | 4.0 ± 6.0 | 0.07817 |
| Mn17_174a | FAT | H456 | 21 Jun 2017 | Canada | adult (pregnant) | F | 177 | 1034.0 ± 1.5 | 20.7 ± 4.3 | 0.07148 |
| Mn17_174b | WIL | H854 | 21 Jun 2017 | Canada | adult | M | 69 | 1035.2 ± 2.9 | 11.7 ± 6.1 | 0.07194 |
| Mn17_178a | FRI | H748 | 25 Jun 2017 | Canada | adult (pregnant) | F | 208 | 1035.0 ± 1.4 | 20.4 ± 2.1 | 0.07907 |
| Mn17_178c | PSE | H008 | 24 Jun 2017 | Canada | adult | F | 215 | 1041.3 ± 2.1 | 1.6 ± 3.7 | 0.06810 |
| Mn17_180a | SIA | H007 | 27 Jun 2017 | Canada | adult | M | 133 | 1037.1 ± 2.2 | 16.0 ± 3.5 | 0.06968 |
| Mn17_180b | RAL | H777 | 29 Jun 2017 | Canada | adult | F | 92 | 1043.6 ± 2.9 | 20.5 ± 3.9 | 0.07399 |
| Mn17_184a | EYE | — | 3 Jul 2017 | Canada | adult | F | 103 | 1038.6 ± 2.6 | 17.9 ± 3.9 | 0.06846 |
| Mn17_186b | STL | H152 | 5 Jul 2017 | Canada | adult | M | 101 | 1038.8 ± 3.0 | 6.1 ± 4.5 | 0.06706 |
| Mn17_186c | SPI | H151 | 5 Jul 2017 | Canada | adult | M | 134 | 1037.6 ± 2.0 | 10.1 ± 3.4 | 0.07619 |
| Mn17_186d | FOS | — | 5 Jul 2017 | Canada | adult | F | 68 | 1031.8 ± 2.2 | 20.3 ± 4.6 | 0.07021 |
| Mn17_190a | MAN | H584 | 7 Jul 2017 | Canada | adult (lactating) | F | 277 | 1035.7 ± 1.7 | 14.4 ± 2.6 | 0.06477 |
| Mn18_013a | ROL | — | 13 Jan 2018 | Norway | juvenile | F | 41 | 1042.0 ± 1.7 | 15.3 ± 2.5 | 0.08406 |

[a]F, M and U of sex column: female, male, unknown sex, respectively.

Using these independent non-invasive methods, we detected expected seasonal changes in body condition during the feeding season. Body condition (i.e. lipid stores) improved during the feeding season for males and females, all age classes and reproductive statuses, as indicated by uTBD declining. Below, we discuss how the body condition of tagged animals varied with sex, reproductive status and location, and how divers may balance energy accumulation with efficient swimming locomotion.

## (a) Seasonal changes in underlying tissue body density

The uTBD decreased during the feeding season ($-2.7$ kg m$^{-3}$ per 100 days) in both feeding areas across sex, age class and reproductive status, indicating successful foraging, which improved body condition. Similar trends were reported by direct measure studies in Iceland: seasonal increased blubber thickness and/or posterior girths of fin whales (*Balaenoptera physalus*) of almost all age classes [49] and linear increases in blubber volumes of mature and pregnant minke whales (*Balaenoptera acutorostrata*) [50]. Narazaki *et al.* [33] repeatedly sampled an adult male (H607) that was also analysed here, and found that its TBD decreased from 1037.0 to 1031.2 kg m$^{-3}$ over 40 days of the feeding season, resulting from the accumulation of lipid stores. The proportion of lipid content ($P_{lipid}$) of the corresponding tissue densities would be 36.3% and 39.0%, based on extrapolation from elephant seals [31,34]. Similarly, the uTBD of resting females in Norway decreased from 1043.2 to 1036.5 kg m$^{-3}$ over 250

days, equivalent to $P_{lipid}$ of 33.4% and 36.5%, respectively. However, converting a specific TBD to an accurate lipid-store content value is only possible if the actual density of lipids and non-lipid tissues is known for humpback whales.

## (b) Body condition in relation to reproductive status

Our study indicated that the lipid-store body condition of humpback whales was closely associated with their reproductive status. The hierarchical model estimated uTBD of pregnant females to be $-3.5$ kg m$^{-3}$ lower than in resting (non-pregnant, non-lactating) females indicating that pregnancy is associated with increased lipid stores. Body condition of pregnant females is expected to be related to reproductive success via improved growth of their offspring: indeed, the growth rate of southern right whale calves was positively correlated with the rate of loss in maternal body volume during the breeding season [28]. In fact, the uTBD of lactating females was higher (+5.8 kg m$^{-3}$) than that of adult females and males, indicating that lactation is associated with decreased lipid stores. The uTBD of lactating females decreased during the feeding season (i.e. increasing lipid store). However, their uTBD remained higher than that of adult males and females throughout the feeding season. These findings indicate that whale mothers might not fully recover the energy expenditure, including lactation demands, incurred during their migration back to the feeding grounds. It might be possible that calving interval would increase if lactating females cannot fully replenish their

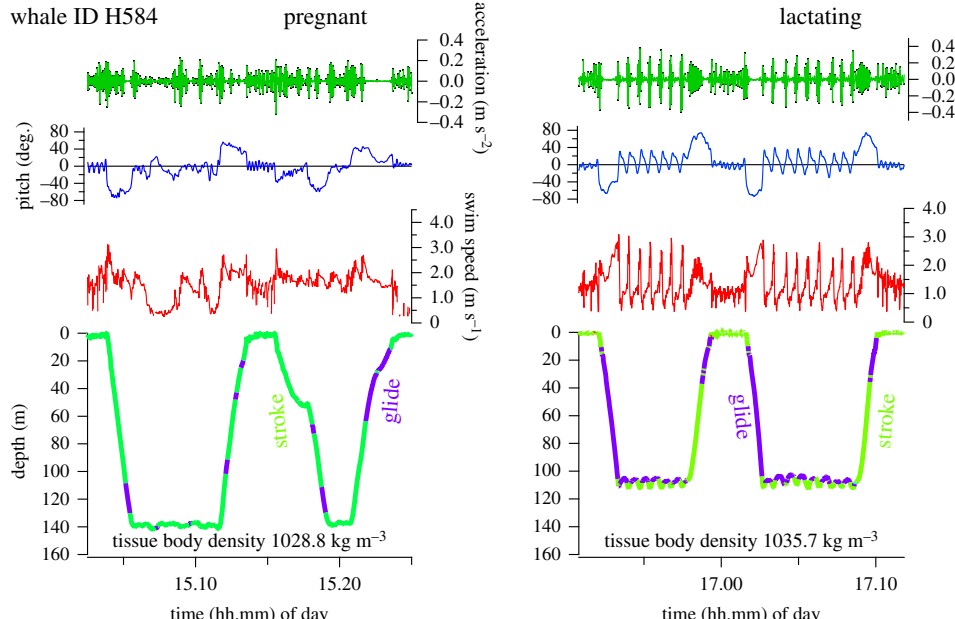

**Figure 3.** Changes in gliding patterns corresponded with tissue body density of the same individuals (ID H584). High values of dorsoventral accelerations indicate periods of fluke-strokes (dive depth in green), while low values indicate gliding periods (dive depth in purple). (Online version in colour.)

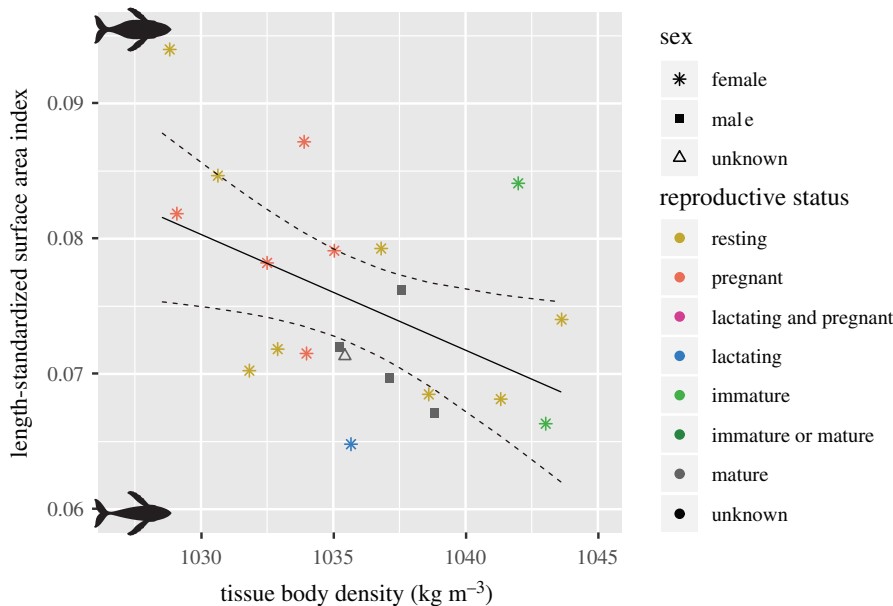

**Figure 4.** Animals with a large length-standardized surface area index show lower estimated TBD from hydrodynamic analyses (see table 1 for detailed information of each individual). The linear regression line is shown in black; 95% confidence intervals are shown as dashed black lines. (Online version in colour.)

energy reserves during the year of lactation. Lactation energy expenditure seems to be eventually recovered as our results showed that uTBD of resting females (1037.4. kg m$^{-3}$) was as low as that of males (1037.2. kg m$^{-3}$).

Although inter- and intra-annual changes to environmental conditions could affect the foraging success of humpback whales, measuring the same individuals tagged multiple times over the course of a single feeding season or over multiple years generated similar trends. Whale H584 had higher TBD during lactation (1035.7 kg m$^{-3}$) than during pregnancy (1028.8 kg m$^{-3}$), suggesting loss in lipid stores during lactation. H002's TBD was higher when resting (1050.9 kg m$^{-3}$) than during pregnancy (1027.9 kg m$^{-3}$). Anatomical measurements from whaling operations show that pregnant female fin whales had the highest increase in energy stores during the feeding season among sex, age

and reproductive classes [1,30]. Pregnant females increased their body mass by 26% and the total energy content of the body by nearly 80% [4]. Although TBD does not directly reflect body mass changes, lower TBD indicates an increasing proportion of lipid stores.

## (c) Geographical variation in body condition and buoyancy

Most Atlantic humpback whales are thought to breed in the West Indies during winter [43], thousands of kilometres from their summer feeding grounds [25,26]. These whales exhibit high maternally directed site fidelity with negligible interchange among groups [25,26]. Photo-identification studies indicate some whales use breeding grounds in Cape Verde [51], although none of the whales tracked from

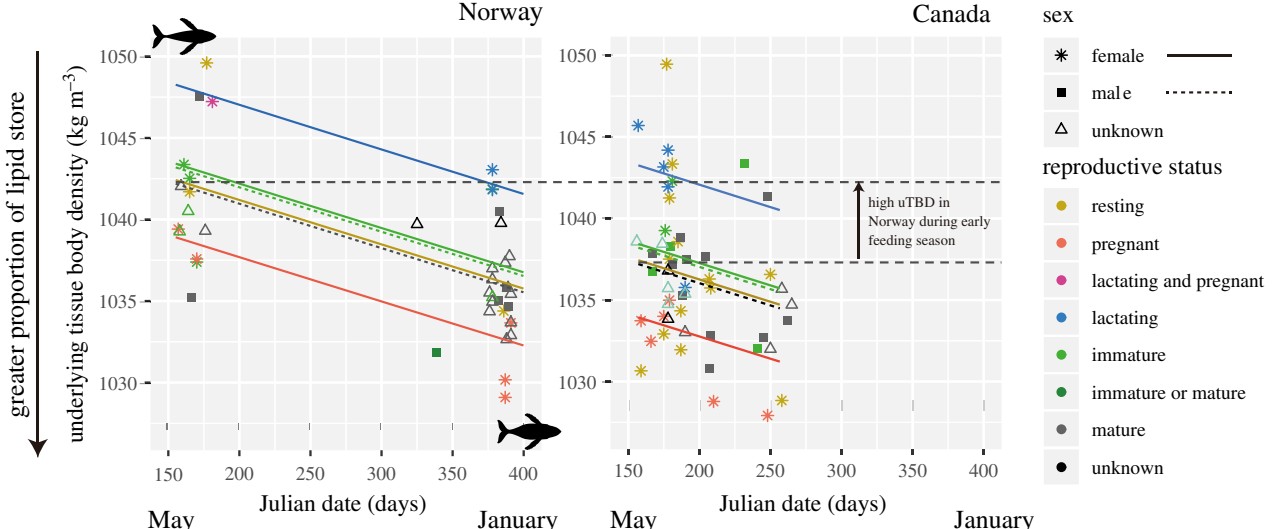

**Figure 5.** Underlying TBD (uTBD) across sex, age classes and reproductive status in Norway and Canada. Each symbol shows individual uTBD estimated from either TBD and/or LSSAI data, coloured by its reproductive status. Both dashed and solid lines indicate decreasing uTBD over feeding seasons predicted by the model (see details in Material and methods). The black up arrow shows differences of uTBD between Norway and Canada during the early feeding season. (Online version in colour.)

Norway have migrated there (https://uit.no/prosjekter/prosjekt?p_document_id=504905).

Monitoring migration routes using satellite tags revealed that some mother-calf pairs of North Atlantic humpback whales require one to two months to migrate to the Gulf of St Lawrence, while other pairs require two to three months to migrate to Norway and Iceland [26]. Breeding areas (West Indies or Cape Verde Islands) of Norwegian whales are, therefore, likely to be 2–3 times further away (one-way trip distance, approx. 8500–9500 km) from feeding areas than those of Canadian whales, resulting in longer fasting times and migration swimming costs. The uTBD of Norwegian whales was higher (indicating relatively lower lipid stores) than that of Canadian whales (+5.0 kg m$^{-3}$) in the early feeding season. The longer migration may explain the proportionally lower lipid stores in Norwegian whales early in the feeding season.

The decline in uTBD during the feeding season might be subject to a trade-off between energy accumulation and loco-motion cost. This is because large amounts of low-density lipids could make whales excessively positively buoyant, increasing locomotion costs of the round-trip from/to depth, and horizontal swimming [19,23,31]. Norwegian whales, for example, might be predicted to accumulate more low-density lipid stores because of their longer migration using coastal 'hotspots' rich in overwintering herring as feeding stopovers during their migration towards the breeding grounds in the south. However, uTBD later in the feeding season was similar in both areas, remaining higher than that of seawater. Thus, the tissue of humpback whales did not typically become positively buoyant, even during the late feeding season. Similarly, negative tissue buoyancy has been found for other cetaceans (1030.0 ± 0.8 kg m$^{-3}$ for sperm whales [21]; 1031.5 ± 1.0 kg m$^{-3}$ for northern bottlenose whales [32], 1038.8 ± 1.6 kg m$^{-3}$ for long-finned pilot whales [20], 1029.8 kg m$^{-3}$ for one beaked whale *Ziphius cavirostris*, K. Aoki 2010, unpublished data).

Neutral buoyancy minimizes round-trip locomotion cost when diving from/to foraging depth, and locomotion cost during horizontal swimming [19,23]. Neutral buoyancy

may offer a particular advantage while feeding at depth, as the whales would not have to overcome positive or negative buoyancy and therefore feed more efficiently. Physostomous fishes achieve neutral buoyancy by adjusting gas volume in the swim bladder, reducing locomotion costs, including the cost of maintaining a particular depth [52]. The adjustment of diving air volume, together with negative tissue buoyancy in humpback whales, can yield neutral buoyancy overall at a shallow swimming depth where gases are not highly compressed. Yet more lipid store than we observed would lead to positive buoyancy that gas stores would only make more extreme. Although greater lipid store body condition may provide a larger energetic buffer prior to migration, maintaining negative tissue buoyancy might drive the target range of TBD to enable efficient migration.

## (d) Conclusion and future directions

Effective methods for measuring body condition enable evaluation of the fitness consequences of changing environmental conditions and prey availability in the Anthropocene [1]. A key advantage of the TBD approach is that it provides a quantitative, numerical estimate of total lipid store body condition, along with estimates of drag and diving gas volume [32]. While body density correlates strongly with total body lipid-store content in mammals [53], conversion of TBD to a specific lipid : lean mass ratio will only be possible once the precise density of lipids and non-lipid tissues is known [34].

Our results confirm that TBD within a given species provides a relative index of body condition across individuals (and changes over time in repeat-sampled individuals). On-board implementation of the body density algorithm in a longer-duration telemetry tag could enable longitudinal tracking of the body condition of individual whales. Tracking changes in lipid stores using tags allowed the resource acquisition and diving energetics of elephant seals to be quantified [34]. Replicating such studies with other diving vertebrates could identify high-quality foraging areas, and quantify energy-store impacts of both natural and anthropogenic disturbances on the body condition of individuals. By contrast,

aerial photogrammetry using UAVs is less invasive than tagging, and many whales can be photographed efficiently [27]. This study provided an independent mechanistic confirmation of the photogrammetry approach, supporting its use as a state-of-the-art technique for instantaneous sampling of cetacean body condition. For identifiable resident species, longitudinal measures of body condition are possible using photogrammetry [28]. For poorly individually marked or less predictable wide-ranging taxa, the tag-based body density method may be most effective for longer-term longitudinal tracking of individuals. Our study demonstrates the benefit of using a combination of methods to estimate body condition when possible, and adds confidence that these two independent methods do effectively estimate the lipid-store body condition of free-ranging cetaceans.

Ethics. The research protocol was approved by the Animal Welfare and Care Committee Approval of the University of St Andrews, UK. The fieldwork in the Gulf of St Lawrence, Canada was performed under the Research permits issued by the Department of Fisheries and Oceans, Canada (scientific fishing license QUE04-B-2011, QUE02-C-2012, QUE-MMO01-2016, QUE-LEP-001-2017) in compliance with ethical and local use of animals in experimentation. Fieldwork in Norway was carried out under a permit issued by the Norwegian Food Health Authority (FOTS ID 8165).

Data accessibility. The data are provided as electronic supplementary material.

Authors' contributions. K.A. participated in the preparation of fieldwork, carried out analysis of tag data, participated in the design of the study and drafted the manuscript; S.I. carried out the statistical analyses and collected field data; C.B. and P.P. carried out data analysis of overhead images and collected field data. L.M.M.L., T.N., T.I., P.J.W. and Y.A. collected field data and carried out analysis of tag data; J.K. collected field data and carried out analysis of biopsy samples. C.R., M.B. and R.S. collected field data and coordinated the field study. K.S. participated in the preparation of fieldwork. A.H. and P.J.O.M. conceived, designed and coordinated the study. All authors edited the manuscript and agree to be held accountable for the work performed therein.

Competing interests. The authors have no conflicts of interest directly relevant to the content of this article.

Funding. This work was supported by the SERDP award (grant no. RC-2337). Additional support for fieldwork was provided by the US Office of Naval Research under the 3S collaborative research project. Analysis and writing were partly supported by the Grant-in-Aid for Young Scientists B (grant no. 17K12813), The Mitsui & Co. Environment Fund (grant no. R16-0044), JSPS Bilateral Open Partnership Joint Research Projects and NERC National Capability funding.

Acknowledgements. We thank L. Kleivane, M. Guilpin and all staff and volunteers from the Mingan Island Cetacean Study for collecting field data. H. Konno helped make tag equipment. Our laboratory members provided many helpful suggestions.

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
