## [Reviewer comments · Proceedings of the Royal Society B: Biological Sciences]

Review History

RSPB-2020-0861.R0 (Original submission)

Review form: Reviewer 1

Recommendation

Major revision is needed (please make suggestions in comments)

Scientific importance: Is the manuscript an original and important contribution to its field?

Good

General interest: Is the paper of sufficient general interest?

Good

Quality of the paper: Is the overall quality of the paper suitable?

Good

Is the length of the paper justified?

Yes

Should the paper be seen by a specialist statistical reviewer?

No

Do you have any concerns about statistical analyses in this paper? If so, please specify them explicitly in your report.

No

It is a condition of publication that authors make their supporting data, code and materials available - either as supplementary material or hosted in an external repository. Please rate, if applicable, the supporting data on the following criteria.

Is it accessible?

No

Is it clear?

N/A

Is it adequate?

No

Do you have any ethical concerns with this paper?

No

Comments to the Author

The authors use a robust data set of tag and UAV data to determine various metrics of body condition in humpback whales on foraging grounds, and evaluate how each data set can be used to effectively estimate and validate measures of body condition. The authors have an excellent data set comprising whales of different age, sex, and reproductive class. I have several specific comments that are listed below. I am curious as well about how/if differences in body condition may be affected by different environmental and prey conditions and how these could affect the results presented.

As a study to show that tags and UAV can provide cross-validation of changes in body density the authors do a very good job and their data are quite outstanding. However, taking the study further to discuss why observed changes in body density occur and how to account for these is less well developed. If the main objective is to compare and validate that body density can be measured, relatively, from tag and UAS data, this was accomplished. As there is less information on the application of this method in other animals I would urge the authors to try and consider ways to make their findings attractive to other systems or types of animals or further develop the reasons (likely environmental and behavioral) that lead to observed differences among different life history stages.

Methods: which tagged whales were flown over? it is hard to find information on this. As well, the image in figure 2 does not seem to be directly overhead and if this is an example of a 'good' image I would have concerns about estimates in body condition. More detailed information on how pixels from different altitudes affect precision of measures is needed.

It would be useful to note in the text during which years and periods the UAS data was collected

line 385: i think that this statement has been validated previously and should include those references.

line 425: not sure what this last sentence adds to the argument

line 428: i would be cautious with this given the possibility of inter annual changes in environmental conditions that could easily affect animal growth and health, as could the time during the feeding season when measurements were taken

line 456: not having read this cited paper, is the timing of migration true for all whales of all age classes? i believe there is a well structured temporal progression of migration of different age/life history classes.

paragraph of line 462: as cost of transport is low for these whales, the amount of time whales are simply not feeding is where energetic changes occur. so is there any evidence that whales feeding in norway spend less time on feeding grounds than those in Canada? As well, differences in prey type will affect energy gain between feeding groups and could influence body condition.

line 477: it is known from other studies that humpback whales late in feeding seasons preferentially target shallow prey rather than diving to depth and requiring more energy to be expended. However, the fact that the whales are in fact in better body condition later in the season speaks to that whatever strategy they are using is working and that the costs associated with diving as a positively buoyant animal does not really affect these whales.

line 486: how could prey availability affect this?

line 492: is there any reference for this statement? I am not sure I believe this given that marine mammals dive to all kinds of depths feeding during all times of year across body sizes successfully.

line 502: think you mean wintering grounds? As well, if you are going to discuss diving air volume it would be helpful to know how much whales can regulate this during diving as it is something that animals can actually manipulate.

Review form: Reviewer 2

Recommendation

Accept with minor revision (please list in comments)

Scientific importance: Is the manuscript an original and important contribution to its field?

Good

General interest: Is the paper of sufficient general interest?

Good

Quality of the paper: Is the overall quality of the paper suitable?

Good

Is the length of the paper justified?

Yes

Should the paper be seen by a specialist statistical reviewer?

Yes

Do you have any concerns about statistical analyses in this paper? If so, please specify them explicitly in your report.

No

It is a condition of publication that authors make their supporting data, code and materials available - either as supplementary material or hosted in an external repository. Please rate, if applicable, the supporting data on the following criteria.

Is it accessible?

Yes

Is it clear?

Yes

Is it adequate?

Yes

Do you have any ethical concerns with this paper?

No

Comments to the Author

General comments

This is a really nice and robust study, combining two novel approaches to assess body condition in baleen whales. The findings are very convincing and fits previous knowledge of seasonal variation in body condition, differences between reproductive classes etc. I only have some minor comments (see specific comments below). The methods section could benefit from some more details about the UAV measurements, the selection of images and the analyses. Also the result section could provide some more details (see comments below). While the paper is very useful for marine mammal research, perhaps better links to terrestrial mammal could be made in the intro and discussion, to make the paper more interesting to the broader readership of Proceedings of the Royal Society B. Overall though, the study is really nice and will make a very valuable contribution to this field.

Specific comments

Lines 29-30: This statement is only true for animals that build up energy reserves. Not all animals do, so please revise.

Lines 30-31: Again, not necessarily. This is a very broad statement, which refers to all animals. Be more specific if you are referring to marine mammals.

Intro: The intro is good and well structured, but perhaps a bit too focused on marine mammals. The first paragraph talks about "animals" in general, but is very specific to marine mammals, with nearly all references also being marine mammal studies. Could you try to make this one a bit broader, focusing perhaps on mammals in general, but also use some examples from the terrestrial world?

Lines 50-52: Again, this is too generic. Not all animals need to do this. Also there are no references.

It would be good to provide a definition of body condition early on in the introduction.

Lines 52-53: Again, too generic. It seems like you are referring to whales in the abstract and intro, but just write animal. If so, please change to marine mammals.

Line 60: Change "reserve energy" to "energy reserves"

Line 65: Can you provide an estimate for lipid density, or at least a range?

Line 66: Same here, can you provide an estimate for lean tissue, or range?

Lines 66-68: Not alone. The amount of air in the lungs of the animal should also affect this.

Lines 70-71: Change “animals” to “marine mammal” since all the references refers to this.

Lines 84-86: You talk about baleen whale migration and reference a seal paper? Please change it.

Lines 88-91: Calf growth rates are also affected by maternal body condition. Check out Christiansen et al. 2018 MEPS.

Lines 115-116: Add “qualitative” between “Visual” and “assessment”. The word “effective” is a bit odd here. What do you mean with it? Its quick? But is it accurate?

Line 125: Add latin name for southern right whale: *Eubalaena australis*.

Lines 137-140: I would remove these sentences and just embed some of the info, e.g. the distance, into the previous sentence.

Lines 140-146: I think this sentence deviates from the main aim of your study. I think this can still be discussed in the discussion section, but should not be part of the intro.

Material and methods

Line 168: The DJI Phantom 4 was released in 2016 I believe. Does that mean that UAV photogrammetry was not performed during the earlier years of sampling?

Lines 172-175: How did the CTD contribute to this study?

Lines 207-208: Please provide a justification for using the compressibility of a northern bottlenose whale for a humpback whale. Is the tissue of a deep diving cetacean comparable to that of a relatively shallow diving humpback?

Lines 240-245: This approach is very similar to Christiansen et al. 2016 Ecosphere, and should be acknowledged. The only difference is that you accounted for body length straight away, whereas Christiansen et al. 2016 did it in a linear model.

Lines 249-251: Why did you only include the surface area from segment 7 (35% BL from rostrum) until 17 (85% BL from rostrum). The latter makes sense, but the start should be at 25%, since that's just behind the eyes, where baleen whales can change quite a lot in width. Alternatively, if you have measurements showing that that is not the case, please provide them to support your cut-off point.

In general, the UAV approach needs more details. How did you select your images to be of adequate quality for photogrammetry? The supplementary materials lists the criteria's you used, but not how they were graded and what were deemed to be “good enough” to be included in the analysis. See the supplementary material of Christiansen et al. (2018 MEPS) for a very detailed description of how UAV images can be graded to assess body condition in baleen whales.

You also need to provide some measure of error for your UAV measurements. Phantom 4s uses quite a wide angle lens, which is likely to result in quite strong edge effects (distortion). Did you account for this by correcting your image? How large were these measurement errors? Some measurements of a known sized object on land (or sea ideally) could help provide these errors and help determine which images (depending on the position of the animal in the photo) are of good enough quality to be used.

Results

First paragraph: Could you please provide a bit more details on the animals you sampled, including their age/size class, body length (if known) etc. This is especially interesting for the 20 whales that you both measured with the UAV and tagged. What range of values did this cover, in terms of body condition (from UAV) etc.

Lines 316-320: Please provide some more details about these findings. What was the effect of these variables on body condition?

Line 378: You are not really using “width-to-length” ratios, but rather surface area to length. That is an important difference, since your metric is captures more of the animals body shape and size. Don’t sell yourself short□ This is a good metric.

Lines 425-426: Provide Latin names for minke, fin and blue whale and for bottlenose dolphin and spotted dolphin.

Lines 432-438: I think you can be a bit more explicit here in terms of the implications of your findings for the tissue composition of humpback whales. That fatter (larger surface area to length) animals have lower TBD shows that the “area” gained during the feeding season is mostly made up of lipids, and hence stored energy, rather than muscle and other heavier tissues. This is very interesting, and should be emphasized I think. Perhaps look more into the whaling literature as well and see how the lipid concentration and blubber thickness of different species changes though the feeding season (this should be available for minke, fin and sei whales at least). These are really cool results, so make sure to highlight the importance of them.

Lines 451-461: There is a lot of unnecessary detail here. Try to shorten this a bit and get faster to the point you are trying to make (that the migratory distance and consequent timing until returning to the feeding grounds) differs between your two study populations.

Line 473: Add “(increased lipid stores)” after “body condition”.

Lines 482-494: It is interesting that the whales seem to end the feeding season in both areas at the same tissue density. While you discuss the benefits of not being too buoyant while feeding, could this also influence migration cost and breeding somehow? Could the locomotion cost during migration be increased in the whales are too buoyant, or could overheating be an issue on the warmer feeding grounds if the whales are too fat when they arrive?
Supplementary materials

Line 77: Change “crewless” to “unmanned”.

Lines 110-114: Please provide additional information on how you evaluated this. Did you grade each image (scored it) based on these criterias? If so, what cut-off scores did you use? You mention “angel” to individual? Wasn’t all the photos taken from straight above (with the camera facing down at a 90 degree angle)? If not, that can seriously impact your measurement errors.

Did your R script account for curved (in the horizontal plane) animals? If not, this could introduce substantial errors in your measurements.

Decision letter (RSPB-2020-0861.R0)

08-Jun-2020

Dear Dr Aoki:

I am writing to inform you that your manuscript RSPB-2020-0861 entitled "Aerial photogrammetry and tag-derived tissue density reveal patterns of lipid-store body condition of humpback whales on their feeding grounds" has, in its current form, been rejected for publication in Proceedings B.

This action has been taken on the advice of referees, who have recommended that substantial revisions are necessary. With this in mind we would be happy to consider a resubmission, provided the comments of the referees are fully addressed. However please note that this is not a provisional acceptance.

Please find below the comments made by the referees, not including confidential reports to the Editor, which I hope you will find useful. As you will see, while there is a consensus that your manuscript has significant potential, there are various substantive issues, relating to assumptions and analyses, that require your careful attention. Additionally, and as emphasised by the Board Member, for us to consider your manuscript further, it will require a more convincing case demonstrating the expected breadth of interest to our diverse readership. If you do choose to resubmit your manuscript, please upload the following:

Sincerely,
Professor Gary Carvalho
mailto: proceedingsb@royalsociety.org

Associate Editor
Board Member: 1
Comments to Author:

Your article was reviewed by two individuals familiar with the topic. While both referees thought you had an excellent data set and analysis was sound, at least one referee felt that the presentation and interpretation of your results needed work. Your work is fundamentally solid, but the implications of your work need to be better developed to make the paper of interest to the broad readership of ProcB.

Reviewer(s)' Comments to Author:

Referee: 1

Comments to the Author(s)

The authors use a robust data set of tag and UAV data to determine various metrics of body condition in humpback whales on foraging grounds, and evaluate how each data set can be used to effectively estimate and validate measures of body condition. The authors have an excellent data set comprising whales of different age, sex, and reproductive class. I have several specific comments that are listed below. I am curious as well about how/if differences in body condition may be affected by different environmental and prey conditions and how these could affect the results presented.

As a study to show that tags and UAV can provide cross-validation of changes in body density the authors do a very good job and their data are quite outstanding. However, taking the study further to discuss why observed changes in body density occur and how to account for these is less well developed. If the main objective is to compare and validate that body density can be measured, relatively, from tag and UAS data, this was accomplished. As there is less information on the application of this method in other animals I would urge the authors to try and consider ways to make their findings attractive to other systems or types of animals or further develop the reasons (likely environmental and behavioral) that lead to observed differences among different life history stages.

Methods: which tagged whales were flown over? it is hard to find information on this. As well, the image in figure 2 does not seem to be directly overhead and if this is an example of a 'good' image I would have concerns about estimates in body condition. More detailed information on how pixels from different altitudes affect precision of measures is needed.

It would be useful to note in the text during which years and periods the UAS data was collected

line 385: i think that this statement has been validated previously and should include those references.

line 425: not sure what this last sentence adds to the argument

line 428: i would be cautious with this given the possibility of inter annual changes in environmental conditions that could easily affect animal growth and health, as could the time during the feeding season when measurements were taken

line 456: not having read this cited paper, is the timing of migration true for all whales of all age classes? i believe there is a well structured temporal progression of migration of different age/life history classes.

paragraph of line 462: as cost of transport is low for these whales, the amount of time whales are simply not feeding is where energetic changes occur. so is there any evidence that whales feeding in norway spend less time on feeding grounds than those in Canada? As well, differences in prey type will affect energy gain between feeding groups and could influence body condition.

line 477: it is known from other studies that humpback whales late in feeding seasons preferentially target shallow prey rather than diving to depth and requiring more energy to be expended. However, the fact that the whales are in fact in better body condition later in the season speaks to that whatever strategy they are using is working and that the costs associated with diving as a positively buoyant animal does not really affect these whales.

line 486: how could prey availability affect this?

line 492: is there any reference for this statement? I am not sure I believe this given that marine mammals dive to all kinds of depths feeding during all times of year across body sizes successfully.

line 502: think you mean wintering grounds? As well, if you are going to discuss diving air volume it would be helpful to know how much whales can regulate this during diving as it is something that animals can actually manipulate.

Referee: 2

Comments to the Author(s)

General comments

This is a really nice and robust study, combining two novel approaches to assess body condition in baleen whales. The findings are very convincing and fits previous knowledge of seasonal variation in body condition, differences between reproductive classes etc. I only have some minor comments (see specific comments below). The methods section could benefit from some more details about the UAV measurements, the selection of images and the analyses. Also the result section could provide some more details (see comments below). While the paper is very useful for marine mammal research, perhaps better links to terrestrial mammal could be made in the intro and discussion, to make the paper more interesting to the broader readership of Proceedings of the Royal Society B. Overall though, the study is really nice and will make a very valuable contribution to this field.

Specific comments

Lines 29-30: This statement is only true for animals that build up energy reserves. Not all animals do, so please revise.

Lines 30-31: Again, not necessarily. This is a very broad statement, which refers to all animals. Be more specific if you are referring to marine mammals.

Intro: The intro is good and well structured, but perhaps a bit too focused on marine mammals. The first paragraph talks about "animals" in general, but is very specific to marine mammals, with nearly all references also being marine mammal studies. Could you try to make this one a bit broader, focusing perhaps on mammals in general, but also use some examples from the terrestrial world?

Lines 50-52: Again, this is too generic. Not all animals need to do this. Also there are no references.

It would be good to provide a definition of body condition early on in the introduction.

Lines 52-53: Again, too generic. It seems like you are referring to whales in the abstract and intro, but just write animal. If so, please change to marine mammals.

Line 60: Change "reserve energy" to "energy reserves"

Line 65: Can you provide an estimate for lipid density, or at least a range?

Line 66: Same here, can you provide an estimate for lean tissue, or range?

Lines 66-68: Not alone. The amount of air in the lungs of the animal should also affect this.

Lines 70-71: Change "animals" to "marine mammal" since all the references refers to this.

Lines 84-86: You talk about baleen whale migration and reference a seal paper? Please change it.

Lines 88-91: Calf growth rates are also affected by maternal body condition. Check out Christiansen et al. 2018 MEPS.

Lines 115-116: Add “qualitative” between “Visual” and “assessment”. The word “effective” is a bit odd here. What do you mean with it? Its quick? But is it accurate?

Line 125: Add latin name for southern right whale: *Eubalaena australis*.

Lines 137-140: I would remove these sentences and just embed some of the info, e.g. the distance, into the previous sentence.

Lines 140-146: I think this sentence deviates from the main aim of your study. I think this can still be discussed in the discussion section, but should not be part of the intro.

Material and methods

Line 168: The DJI Phantom 4 was released in 2016 I believe. Does that mean that UAV photogrammetry was not performed during the earlier years of sampling?

Lines 172-175: How did the CTD contribute to this study?

Lines 207-208: Please provide a justification for using the compressibility of a northern bottlenose whale for a humpback whale. Is the tissue of a deep diving cetacean comparable to that of a relatively shallow diving humpback?

Lines 240-245: This approach is very similar to Christiansen et al. 2016 Ecosphere, and should be acknowledged. The only difference is that you accounted for body length straight away, whereas Christiansen et al. 2016 did it in a linear model.

Lines 249-251: Why did you only include the surface area from segment 7 (35% BL from rostrum) until 17 (85% BL from rostrum). The latter makes sense, but the start should be at 25%, since that's just behind the eyes, where baleen whales can change quite a lot in width. Alternatively, if you have measurements showing that that is not the case, please provide them to support your cut-off point.

In general, the UAV approach needs more details. How did you select your images to be of adequate quality for photogrammetry? The supplementary materials lists the criteria's you used, but not how they were graded and what were deemed to be “good enough” to be included in the analysis. See the supplementary material of Christiansen et al. (2018 MEPS) for a very detailed description of how UAV images can be graded to assess body condition in baleen whales.

You also need to provide some measure of error for your UAV measurements. Phantom 4s uses quite a wide angle lens, which is likely to result in quite strong edge effects (distortion). Did you account for this by correcting your image? How large were these measurement errors? Some measurements of a known sized object on land (or sea ideally) could help provide these errors and help determine which images (depending on the position of the animal in the photo) are of good enough quality to be used.

Results

First paragraph: Could you please provide a bit more details on the animals you sampled, including their age/size class, body length (if known) etc. This is especially interesting for the 20 whales that you both measured with the UAV and tagged. What range of values did this cover, in terms of body condition (from UAV) etc.

Lines 316-320: Please provide some more details about these findings. What was the effect of these variables on body condition?

Line 378: You are not really using “width-to-length” ratios, but rather surface area to length. That is an important difference, since your metric captures more of the animal's body shape and size. Don't sell yourself short □ This is a good metric.

Lines 425-426: Provide Latin names for minke, fin and blue whale and for bottlenose dolphin and spotted dolphin.

Lines 432-438: I think you can be a bit more explicit here in terms of the implications of your findings for the tissue composition of humpback whales. That fatter (larger surface area to length) animals have lower TBD shows that the “area” gained during the feeding season is mostly made up of lipids, and hence stored energy, rather than muscle and other heavier tissues. This is very interesting, and should be emphasized I think. Perhaps look more into the whaling literature as well and see how the lipid concentration and blubber thickness of different species changes though the feeding season (this should be available for minke, fin and sei whales at least). These are really cool results, so make sure to highlight the importance of them.

Lines 451-461: There is a lot of unnecessary detail here. Try to shorten this a bit and get faster to the point you are trying to make (that the migratory distance and consequent timing until returning to the feeding grounds) differs between your two study populations.

Line 473: Add “(increased lipid stores)” after “body condition”.

Lines 482-494: It is interesting that the whales seem to end the feeding season in both areas at the same tissue density. While you discuss the benefits of not being too buoyant while feeding, could this also influence migration cost and breeding somehow? Could the locomotion cost during migration be increased in the whales are too buoyant, or could overheating be an issue on the warmer feeding grounds if the whales are too fat when they arrive?
Supplementary materials

Line 77: Change “crewless” to “unmanned”.

Lines 110-114: Please provide additional information on how you evaluated this. Did you grade each image (scored it) based on these criteria? If so, what cut-off scores did you use? You mention “angel” to individual? Wasn't all the photos taken from straight above (with the camera facing down at a 90 degree angle)? If not, that can seriously impact your measurement errors.

Did your R script account for curved (in the horizontal plane) animals? If not, this could introduce substantial errors in your measurements.

Author's Response to Decision Letter for (RSPB-2020-0861.R0)

See Appendix A.

RSPB-2020-2307.R0

Review form: Reviewer 2

Recommendation

Accept with minor revision (please list in comments)

Scientific importance: Is the manuscript an original and important contribution to its field?

Excellent

General interest: Is the paper of sufficient general interest?

Good

Quality of the paper: Is the overall quality of the paper suitable?

Excellent

Is the length of the paper justified?

Yes

Should the paper be seen by a specialist statistical reviewer?

No

Do you have any concerns about statistical analyses in this paper? If so, please specify them explicitly in your report.

No

It is a condition of publication that authors make their supporting data, code and materials available - either as supplementary material or hosted in an external repository. Please rate, if applicable, the supporting data on the following criteria.

Is it accessible?

N/A

Is it clear?

N/A

Is it adequate?

N/A

Do you have any ethical concerns with this paper?

No

Comments to the Author

General comments

This is the second time I have the pleasure to review this very nice study, and again I am very supportive of it being published in Proceedings B after just some minor revisions. The study convincingly validates both the glide-method and the photogrammetry method to measure body condition in cetaceans, and will hence be a valuable contribution to this field. The study uses innovative technologies and analytical methods, and a relatively large sample size (given how difficult it is to obtain paired samples from the same individuals) to support their conclusions. They clearly present their findings in context of earlier work in this area, which hence adds further support to previous work (and vice versa). Minor edits and comments follow below. Overall, a very nice study.

Specific comments

Line 31: Add "energy" before "reserves".

Line 33: Provide latin name for humpback whale.

Lines 51-52: This statement ("accumulate") is only true for animals that build up energy reserves. Not all animals do, so please revise.

Lines 53-54: Again, not necessarily. This is a very broad statement, which refers to all animals. Be more specific if you are referring to marine mammals.

Line 57: Add “generally” before “have”, as there are exceptions to this.

Line 60: Could you not change “dynamic metric” to “parameter” or “variable” instead. Also what do you mean with “individual state”, please be more specific.

Line 75: Baleen whales also store significant amounts of energy in their muscle, as visceral fats, and also in other tissues (e.g. bones). Whaling data shows that muscle and visceral fats plays a very important role in some species (e.g. minke whales). This needs to be mentioned.

Line 97: Add “good” before “predictor”.

Line 98: Refs 26 and 27 does not refer to foetal development, but to calf body condition (ref 26) and growth (ref 27). Add Christiansen et al. 2014. *Functional Ecology* 28: 579-588.

Lines 98-99: Change “body condition, and survival of offspring” to “offspring body condition, growth and survival” to make it clearer that you are referring to the offspring and not adult body condition (since “quantity of energy stored as lipid” is body condition). You can use refs 26 and 27 here.

Line 125: Add “()” around the latin name to be consistent with earlier formatting. Change “recognize” to “identify”.

Lines 130-132: The end of this sentence is strangely worded. Consider revising it to “demonstrating the importance of lipid accumulation during the feeding season for successful breeding”.

Lines 137-139: Good that you are mentioning this, however it would be even better if you also mentioned it earlier in the intro (see comment to line 75 above).

Line 142: Add “feeding” before “grounds”.

Line 145: The distance provided here is confusing. Spell out what it is. Is it the extra distance travelled, the distance of migration one way, or the total round trip distance of the migration.

Lines 146-148: Actually, migration in baleen whales is considered to be relatively cheap (see Braithwaite et al. 2015. *Conservation Physiology* 3.1: cov001), with the main costs coming from the number of days it takes the whales to migrate (costs of maintenance during those days) rather than from swimming. So the extra costs for the Norwegian whales would come from the additional days it takes them to reach their breeding grounds compared to Canadian whales. Also why would Norwegian whales have lower energy stores at the start of the feeding season? I would have thought that humpback whales, irrespective of their feeding ground, would aim to time their migration so that they arrive back at the start of the feeding season with a given amount of energy (enough to survive and provide a bit of a buffer). To achieve this while still having a long breeding season, the Norwegian whales would have to feed for longer (or at a faster rate, although it makes little sense why whales wouldn't always be feeding at their fastest rate) and build up larger reserves before departing the feeding grounds, as you propose.

Lines 159-162: Your dates of fieldwork differs a lot between your two study sites, which makes it difficult to test your hypotheses relating to arrival and departure body condition. How did you account for this difference in sampling period?

Line 165: Change “the” to “a”.

Line 239: Write out what sections 7 to 17 represents as % body length from rostrum, e.g. (35-85% body length from the rostrum).

Line 273: Why are you reporting photos of non-tagged animals? Did you use these for your analyses, or only the photos of the tagged whales? If the latter, then your true sample size was 21 whales.

Line 278: Please change “The global species” to something more appropriate. You only measured two populations of humpback whales in the North Atlantic. Extrapolating this to all humpback whale populations is not appropriate, as there could be morphological and anatomical differences between populations (e.g. the northern hemisphere and the southern hemisphere populations).

Lines 295-300: This is confusing. First you report that you obtained repeated measurements from three whales, but then state “these two individuals”. What happened with H607?

Lines 311-314: This is a very nice finding!

Line 318-322: Very nice!

Line 351: Replace “are” with “have been”.

Line 376: Provide latin name for fin whale.

Lines 397-399: This makes a lot of sense if you consider that humpback whales normally have inter-calving intervals of 2-3 years. Perhaps you could expand on this sentence and speculate on what this means for the inter-calving intervals of humpback whales.

Line 401: Change “animals” to “humpback whales”.

Lines 426-429: See comment above (Line 145 and Lines 146-148).

Lines 431-432: It would be nice to see you expand on this and discuss what it actually means for their migratory behaviour. If whales in Norway are able to return with lower body condition stores and still survive and reproduce, why would Canadian whales not do the same, and stay longer on the breeding grounds. Could there be differences in the feeding rates between the two areas that makes it possible for Norwegian whales to use up more energy during the breeding season compared to Canadian whales.

Review form: Reviewer 3

Recommendation

Major revision is needed (please make suggestions in comments)

Scientific importance: Is the manuscript an original and important contribution to its field?

Acceptable

General interest: Is the paper of sufficient general interest?

Acceptable

Quality of the paper: Is the overall quality of the paper suitable?

Acceptable

Is the length of the paper justified?

Yes

Should the paper be seen by a specialist statistical reviewer?

No

Do you have any concerns about statistical analyses in this paper? If so, please specify them explicitly in your report.

No

It is a condition of publication that authors make their supporting data, code and materials available - either as supplementary material or hosted in an external repository. Please rate, if applicable, the supporting data on the following criteria.

Is it accessible?

Yes

Is it clear?

Yes

Is it adequate?

Yes

Do you have any ethical concerns with this paper?

No

Comments to the Author

Aoki et al. estimated the body density (a proxy for body condition) of humpback whales based on tag data combined with hydrodynamic modeling. They cross-checked their estimates against independent estimate of relative fatness of the animals examined by drone images. They applied the method to show some regional, seasonal, and inter-individual differences in body condition in this long-migrating species.

The cross validation of two independent methods using a large dataset (>50 individuals) is an important step forward in monitoring the body conditions of large whales in the wild and understanding underlying factors. The regional and seasonal changes in body condition demonstrated in this study are interesting as they have important implications for the migration strategies of this species. These are two main strength of this paper. On the other hand, I have a major concern about the structure of the paper. It has two component (the method component and ecology component) and the first component is the main one. However, the method for estimating body density based on gliding periods has already been established by a series of studies cited in this paper, including Aoki et al. (2011) that validated the method, although the validation using drone images in large whales is new. For this reason, I believe the main focus of this paper should be shifted to the second, ecology component. Most of the introduction is dedicated to the method component, whereas a large part of the discussion is about the ecology component. This unbalance makes this paper hard to read in the current form.

Specific issues.

Introduction should be condensed, given that the glide method has already been established.

L286-292. Are the values of drag term and air volume important in this study?

L336. The opposite argument is possible. The drone method, which is easier, is good enough for estimating animal body conditions?

L452. Many teleosts have neutral buoyancy to remain stationary in the water column, not for efficient locomotion. Highly active fishes, such as tunas, have negative buoyancy.

Review form: Reviewer 4

Recommendation

Accept with minor revision (please list in comments)

Scientific importance: Is the manuscript an original and important contribution to its field?

Excellent

General interest: Is the paper of sufficient general interest?

Excellent

Quality of the paper: Is the overall quality of the paper suitable?

Excellent

Is the length of the paper justified?

Yes

Should the paper be seen by a specialist statistical reviewer?

No

Do you have any concerns about statistical analyses in this paper? If so, please specify them explicitly in your report.

No

It is a condition of publication that authors make their supporting data, code and materials available - either as supplementary material or hosted in an external repository. Please rate, if applicable, the supporting data on the following criteria.

Is it accessible?

Yes

Is it clear?

Yes

Is it adequate?

Yes

Do you have any ethical concerns with this paper?

No

Comments to the Author

The authors have presented a robust and compelling comparison of two methods for assessing variation in body condition in large cetaceans, a critical metric for understanding prey availability and reproductive success/fitness. I think the authors did a satisfactory job of address the first set of reviewers' comments, in particular with regard to clarifying some UAV methodologies and providing more compelling general context for their study, particularly in the introduction. I only have a few comments to further improve clarity.

Introduction

Line 60-63: Cumbersome run-on sentence, the authors are trying to link several distinct ideas - consider splitting into two or simplifying the wording.

Line 75: "energy reserves" instead of "body reserves"

Line 76-78: List of blubber functions isn't structured properly

Line 91: "a further increase in low-density lipid stores" helps clarify/distinguish this sentence from the previous one.

Line 131-132: move "to successful breeding" to end of sentence.

Line 133-137: while this information is referred to briefly in the discussion, I do not think it is necessary in the introduction and detracts from the flow of the section. I would remove this entirely and incorporate the remainder of the paragraph (starting with "While most baleen whale...") toward the end of the TBD paragraph (Line 100-117)

Results

Lines 266-275: The lengthy lists of number of animals-per-category I find to be an inefficient method for delivering the information. Could all of this information be compiled into a table instead?

Line 421: "whales tracked from Norway have" instead of "Norway has"

Discussion:

Lines 433-458: in this discussion of cetaceans remaining slightly negatively buoyant (though near neutral) to compensate for lung volume, it may be interesting to contrast them with elephant seals, who DO exhibit positive buoyancy but dive with collapsed lungs and therefore don't have to compensate for buoyant gas. Similarly, is there any evidence of deep-diving cetaceans (sperm or beaked whales) that have lower tissue densities since they also would not have the same air volume to counteract?

Decision letter (RSPB-2020-2307.R0)

23-Nov-2020

Dear Dr Aoki:

Your manuscript has now been peer reviewed and the reviews have been assessed by an Associate Editor. The reviewers' comments (not including confidential comments to the Editor) and the comments from the Associate Editor are included at the end of this email for your reference. As you will see, the reviewers and the Editors have raised some concerns with your manuscript and we would like to invite you to revise your manuscript to address them.

To submit your revision please log into <http://mc.manuscriptcentral.com/prsb> and enter your Author Centre, where you will find your manuscript title listed under "Manuscripts with

Decisions." Under "Actions", click on "Create a Revision". Your manuscript number has been appended to denote a revision.

Research ethics:

Use of animals and field studies:

It is a condition of publication that you make available the data and research materials supporting the results in the article (<https://royalsociety.org/journals/authors/author-guidelines/#data>). Datasets should be deposited in an appropriate publicly available repository and details of the associated accession number, link or DOI to the datasets must be included in the Data Accessibility section of the article (<https://royalsociety.org/journals/ethics-policies/data-sharing-mining/>). Reference(s) to datasets should also be included in the reference list of the article with DOIs (where available).

All supplementary materials accompanying an accepted article will be treated as in their final form. They will be published alongside the paper on the journal website and posted on the online figshare repository. Files on figshare will be made available approximately one week before the

accompanying article so that the supplementary material can be attributed a unique DOI. Please try to submit all supplementary material as a single file.

Please submit a copy of your revised paper within three weeks. If we do not hear from you within this time your manuscript will be rejected. If you are unable to meet this deadline please let us know as soon as possible, as we may be able to grant a short extension.

Best wishes,
Professor Gary Carvalho
mailto:proceedingsb@royalsociety.org

Associate Editor Board Member

Comments to Author:

Your ms has been reviewed by two of the referees are mostly satisfied with your revisions, they have a few more suggestions that should be completed before your paper is accepted. The third referee questioned whether your paper was of sufficient general interest, so I would also recommend revising your ms with that in mind.

Reviewer(s)' Comments to Author:

Referee: 3

Comments to the Author(s).

Aoki et al. estimated the body density (a proxy for body condition) of humpback whales based on tag data combined with hydrodynamic modeling. They cross-checked their estimates against independent estimate of relative fatness of the animals examined by drone images. They applied the method to show some regional, seasonal, and inter-individual differences in body condition in this long-migrating species.

The cross validation of two independent methods using a large dataset (>50 individuals) is an important step forward in monitoring the body conditions of large whales in the wild and understanding underlying factors. The regional and seasonal changes in body condition demonstrated in this study are interesting as they have important implications for the migration strategies of this species. These are two main strength of this paper. On the other hand, I have a major concern about the structure of the paper. It has two component (the method component and ecology component) and the first component is the main one. However, the method for estimating body density based on gliding periods has already been established by a series of studies cited in this paper, including Aoki et al. (2011) that validated the method, although the validation using drone images in large whales is new. For this reason, I believe the main focus of this paper should be shifted to the second, ecology component. Most of the introduction is dedicated to the method component, whereas a large part of the discussion is about the ecology component. This unbalance makes this paper hard to read in the current form.

Specific issues.

Introduction should be condensed, given that the glide method has already been established.

L286-292. Are the values of drag term and air volume important in this study?

L336. The opposite argument is possible. The drone method, which is easier, is good enough for estimating animal body conditions?

L452. Many teleosts have neutral buoyancy to remain stationary in the water column, not for efficient locomotion. Highly active fishes, such as tunas, have negative buoyancy.

Referee: 2

Comments to the Author(s).

General comments

This is the second time I have the pleasure to review this very nice study, and again I am very supportive of it being published in Proceedings B after just some minor revisions. The study convincingly validates both the glide-method and the photogrammetry method to measure body condition in cetaceans, and will hence be a valuable contribution to this field. The study uses innovative technologies and analytical methods, and a relatively large sample size (given how difficult it is to obtain paired samples from the same individuals) to support their conclusions. They clearly present their findings in context of earlier work in this area, which hence adds further support to previous work (and vice versa). Minor edits and comments follow below. Overall, a very nice study.

Specific comments

Line 31: Add "energy" before "reserves".

Line 33: Provide latin name for humpback whale.

Lines 51-52: This statement ("accumulate") is only true for animals that build up energy reserves. Not all animals do, so please revise.

Lines 53-54: Again, not necessarily. This is a very broad statement, which refers to all animals. Be more specific if you are referring to marine mammals.

Line 57: Add "generally" before "have", as there are exceptions to this.

Line 60: Could you not change "dynamic metric" to "parameter" or "variable" instead. Also what do you mean with "individual state", please be more specific.

Line 75: Baleen whales also store significant amounts of energy in their muscle, as visceral fats, and also in other tissues (e.g. bones). Whaling data shows that muscle and visceral fats plays a very important role in some species (e.g. minke whales). This needs to be mentioned.

Line 97: Add "good" before "predictor".

Line 98: Refs 26 and 27 does not refer to foetal development, but to calf body condition (ref 26) and growth (ref 27). Add Christiansen et al. 2014. Functional Ecology 28: 579-588.

Lines 98-99: Change "body condition, and survival of offspring" to "offspring body condition, growth and survival" to make it clearer that you are referring to the offspring and not adult body condition (since "quantity of energy stored as lipid" is body condition). You can use refs 26 and 27 here.

Line 125: Add "()" around the latin name to be consistent with earlier formatting. Change "recognize" to "identify".

Lines 130-132: The end of this sentence is strangely worded. Consider revising it to “demonstrating the importance of lipid accumulation during the feeding season for successful breeding”.

Lines 137-139: Good that you are mentioning this, however it would be even better if you also mentioned it earlier in the intro (see comment to line 75 above).

Line 142: Add “feeding” before “grounds”.

Line 145: The distance provided here is confusing. Spell out what it is. Is it the extra distance travelled, the distance of migration one way, or the total round trip distance of the migration.

Lines 146-148: Actually, migration in baleen whales is considered to be relatively cheap (see Braithwaite et al. 2015. Conservation Physiology 3.1: cov001), with the main costs coming from the number of days it takes the whales to migrate (costs of maintenance during those days) rather than from swimming. So the extra costs for the Norwegian whales would come from the additional days it takes them to reach their breeding grounds compared to Canadian whales. Also why would Norwegian whales have lower energy stores at the start of the feeding season? I would have thought that humpback whales, irrespective of their feeding ground, would aim to time their migration so that they arrive back at the start of the feeding season with a given amount of energy (enough to survive and provide a bit of a buffer). To achieve this while still having a long breeding season, the Norwegian whales would have to feed for longer (or at a faster rate, although it makes little sense why whales wouldn't always be feeding at their fastest rate) and build up larger reserves before departing the feeding grounds, as you propose.

Lines 159-162: Your dates of fieldwork differs a lot between your two study sites, which makes it difficult to test your hypotheses relating to arrival and departure body condition. How did you account for this difference in sampling period?

Line 165: Change “the” to “a”.

Line 239: Write out what sections 7 to 17 represents as % body length from rostrum, e.g. (35-85% body length from the rostrum).

Line 273: Why are you reporting photos of non-tagged animals? Did you use these for your analyses, or only the photos of the tagged whales? If the latter, then your true sample size was 21 whales.

Line 278: Please change “The global species” to something more appropriate. You only measured two populations of humpback whales in the North Atlantic. Extrapolating this to all humpback whale populations is not appropriate, as there could be morphological and anatomical differences between populations (e.g. the northern hemisphere and the southern hemisphere populations).

Lines 295-300: This is confusing. First you report that you obtained repeated measurements from three whales, but then state “these two individuals”. What happened with H607?

Lines 311-314: This is a very nice finding!

Line 318-322: Very nice!

Line 351: Replace “are” with “have been”.

Line 376: Provide latin name for fin whale.

Lines 397-399: This makes a lot of sense if you consider that humpback whales normally have inter-calving intervals of 2-3 years. Perhaps you could expand on this sentence and speculate on what this means for the inter-calving intervals of humpback whales.

Line 401: Change "animals" to "humpback whales".

Lines 426-429: See comment above (Line 145 and Lines 146-148).

Lines 431-432: It would be nice to see you expand on this and discuss what it actually means for their migratory behaviour. If whales in Norway are able to return with lower body condition stores and still survive and reproduce, why would Canadian whales not do the same, and stay longer on the breeding grounds. Could there be differences in the feeding rates between the two areas that makes it possible for Norwegian whales to use up more energy during the breeding season compared to Canadian whales.

Referee: 4

Comments to the Author(s).

The authors have presented a robust and compelling comparison of two methods for assessing variation in body condition in large cetaceans, a critical metric for understanding prey availability and reproductive success/fitness. I think the authors did a satisfactory job of address the first set of reviewers' comments, in particular with regard to clarifying some UAV methodologies and providing more compelling general context for their study, particularly in the introduction. I only have a few comments to further improve clarity.

Introduction

Line 60-63: Cumbersome run-on sentence, the authors are trying to link several distinct ideas - consider splitting into two or simplifying the wording.

Line 75: "energy reserves" instead of "body reserves"

Line 76-78: List of blubber functions isn't structured properly

Line 91: "a further increase in low-density lipid stores" helps clarify/distinguish this sentence from the previous one.

Line 131-132: move "to successful breeding" to end of sentence.

Line 133-137: while this information is referred to briefly in the discussion, I do not think it is necessary in the introduction and detracts from the flow of the section. I would remove this entirely and incorporate the remainder of the paragraph (starting with "While most baleen whale...") toward the end of the TBD paragraph (Line 100-117)

Results

Lines 266-275: The lengthy lists of number of animals-per-category I find to be an inefficient method for delivering the information. Could all of this information be compiled into a table instead?

Line 421: "whales tracked from Norway have" instead of "Norway has"

Discussion:

Lines 433-458: in this discussion of cetaceans remaining slightly negatively buoyant (though near neutral) to compensate for lung volume, it may be interesting to contrast them with elephant seals, who DO exhibit positive buoyancy but dive with collapsed lungs and therefore don't have to compensate for buoyant gas. Similarly, is there any evidence of deep-diving cetaceans (sperm

or beaked whales) that have lower tissue densities since they also would not have the same air volume to counteract?

Author's Response to Decision Letter for (RSPB-2020-2307.R0)

See Appendix B.

Decision letter (RSPB-2020-2307.R1)

24-Dec-2020

Dear Dr Aoki

I am pleased to inform you that your manuscript entitled "Aerial photogrammetry and tag-derived tissue density reveal patterns of lipid-store body condition of humpback whales on their feeding grounds" has been accepted for publication in Proceedings B.

Open Access

Paper charges

Sincerely,
Professor Gary Carvalho
Editor, Proceedings B
mailto: proceedingsb@royalsociety.org

Associate Editor:
Board Member
Comments to Author:
I am satisfied that you addressed the comments of the reviewers.

Kagari Aoki, Ph.D

Atmosphere and Ocean Research Institute, The University of Tokyo
5-1-5 Kashiwanoha, Kashiwa City, Chiba Prefecture, 277-8564, JAPAN
TEL: +81-4-7136-6226
E-mail: aokikagari@aori.u-tokyo.ac.jp

22 October 2020

I am pleased to resubmit an original article for publication in *Proceedings of the Royal Society B*, titled “**Aerial photogrammetry and tag-derived tissue density reveal patterns of lipid-store body condition of humpback whales on their feeding grounds.**” The paper was coauthored by Kagari Aoki, Saana Isojunno, Charlotte Bellot, Takashi Iwata, Joanna Kershaw, Yu Akiyama, Lucía Martina Martín López, Christian Ramp, Martin Biuw, René Swift, Paul Wensveen, Patrick Pomeroy, Tomoko Narazaki, Ailsa Hall, Katsufumi Sato, and Patrick J. O. Miller.

This study cross-validated two non-invasive approaches to measure the body condition of humpback whales (tissue body-density estimation using animal-borne tags and shape measurement from overhead photogrammetry images), and applied the data collected to obtain insights on how lipid stores contribute to life-history traits. We believe that our study makes a significant contribution to the published literature because validating these approaches is crucial to interpret indices of body condition, link individual condition to population vital rates, and, ultimately, advance our understanding of the ecology of multiple cetacean species globally.

Based on your letter dated 8 June 2019, we have considerably revised our manuscript by following the suggestions of the reviewers, which we believe has greatly improved it. We provide our responses to the comments of all reviewers below. We thank the editor and reviewers for their constructive comments.

We hope that the revised manuscript is now suitable for the publication in *Proceedings of the Royal Society B*.

Yours sincerely,

Kagari Aoki

Reviewer comments are presented in black font, and our responses are presented in green font.

Reviewer Comments to Author:

Referee: 1

Comments to the Author(s)

The authors use a robust data set of tag and UAV data to determine various metrics of body condition in humpback whales on foraging grounds, and evaluate how each data set can be used to effectively estimate and validate measures of body condition. The authors have an excellent data set comprising whales of different age, sex, and reproductive class. I have several specific comments that are listed below. I am curious as well about how/if differences in body condition may be affected by different environmental and prey conditions and how these could affect the results presented.

As a study to show that tags and UAV can provide cross-validation of changes in body density the authors do a very good job and their data are quite outstanding. However, taking the study further to discuss why observed changes in body density occur and how to account for these is less well developed. If the main objective is to compare and validate that body density can be measured, relatively, from tag and UAS data, this was accomplished. As there is less information on the application of this method in other animals I would urge the authors to try and consider ways to make their findings attractive to other systems or types of animals or further develop the reasons (likely environmental and behavioral) that lead to observed differences among different life history stages.

Thank you for providing useful suggestions. As the referee stated, we focused on validating body condition using two independent methods (UAV and animal-borne tags) in this study. This knowledge is vital towards investigating body condition in relation to changing environmental conditions and prey availability in the future. We have added content in the general introduction and discussion to broaden the appeal of our results to readership of the journal, in relation to these components.

Methods: which tagged whales were flown over? it is hard to find information on this. As well, the image in figure 2 does not seem to be directly overhead and if this is an example of a 'good' image I would have concerns about estimates in body condition. More detailed information on how pixels from different altitudes affect precision of measures is needed.

It would be useful to note in the text during which years and periods the UAS data was collected

We conducted UAV flights in 2016-2018. We added this information to the Materials and Methods. We also added information of the altitude that UAVs were flown in the supplementary material. Because we used relative measures of width versus length, the absolute value of the metrics did not impact our conclusions. Relative values have minimal errors as long as the image is not distorted

We had weak light, especially during winter in Norway, and rarely had flat calm seas. The behaviour of feeding whales is more challenging in these areas because they spend more time at depth than in breeding ground areas, where they tend to remain closer to the surface. Such logging behaviour was rarely observed in our studies on the feeding grounds. Our results contained a degree of noise, as is any empirical measurement, but was sufficient to demonstrate the clear relationship between LSSAI and body density. Using LIDAR as a technology to obtain absolute measurements is advantageous, and we recommend its use in the future. However, this system was not available at the time of our study. The basis of our photogrammetry analysis is well grounded in publications from before LIDAR became more commonly used (e.g. Miller et al., 2012; Mar. Ecol. Prog. Ser. 459, 135-156). We have added detailed information of how we selected usable UAV images in supplements.

line 385: I think that this statement has been validated previously and should include those references.

As requested, we have added the relevant references.

line 425: not sure what this last sentence adds to the argument

Based on the reviewer's comment, we have removed this sentence.

line 428: i would be cautious with this given the possibility of inter annual changes in environmental conditions that could easily affect animal growth and health, as could the time during the feeding season when measurements were taken

Thank you for these useful suggestions. We have revised these sentences to state: "Although inter- and intra-annual changes to environmental conditions could affect the foraging success of animals, measuring the same individuals tagged multiple times over the course of a single feeding season or over multiple years generated similar trends."

line 456: not having read this cited paper, is the timing of migration true for all whales of all age classes? i believe there is a well structured temporal progression of migration of different age/life history classes.

Only a few examples are given in this cited paper. I have added more information on the sex and age/class, stating: "Monitoring migration routes using satellite tags revealed that some mother-calf pairs of North Atlantic humpback whales require 1–2 months to migrate to the Gulf of St. Lawrence, while other pairs require 2–3 months to migrate to Norway and Iceland (Kennedy et al. 2013)."

paragraph of line 462: as cost of transport is low for these whales, the amount of time whales are simply not feeding is where energetic changes occur. so is there any evidence that whales feeding in Norway spend less time on feeding grounds than

those in Canada? As well, differences in prey type will affect energy gain between feeding groups and could influence body condition.

We have inserted text stating “resulting in longer fasting times and migration swimming costs.”

Robust information on the time individuals spend feeding in different areas is not available. We measured tissue body density, which indicates proportional lipid stores. Differences in the type of prey consumed likely impact energy gain; however, it was not possible to evaluate differences among prey items (i.e., different nutrient content of prey) based on our results.

line 477: it is known from other studies that humpback whales late in feeding seasons preferentially target shallow prey rather than diving to depth and requiring more energy to be expended. However, the fact that the whales are in fact in better body condition later in the season speaks to that whatever strategy they are using is working and that the costs associated with diving as a positively buoyant animal does not really affect these whales.

Whatever depth an animal dives, if the buoyancy deviates from neutral buoyancy, additional locomotion is required, incurring energetic costs (Sato et al., 2013, Miller et al., 2012). This cost should increase with distance (dive depth). This extra cost is incurred to stay at the same depth, even when diving depth is shallow. This cost is real, as it is predictable from physics; however, sufficient energy from prey intake was clearly able to cover any increase in locomotion cost related to deviation from neutral buoyancy.

line 486: how could prey availability affect this?

We have revised these sentences to focus on buoyancy: “However, uTBD later in the feeding season was similar in both areas, remaining higher than that of seawater. Thus, the tissue of humpback whale did not typically become positively buoyant, even

during the late feeding season. Similarly, negative tissue buoyancy has been found for other cetaceans ($1030.0 \pm 0.8 \text{ kg m}^{-3}$, sperm whales, Miller et al. 2004; $1031.5 \pm 1.0 \text{ kg m}^{-3}$, northern bottlenose whales, Miller et al. 2016; $1038.8 \pm 1.6 \text{ kg m}^{-3}$, long-finned pilot whales, Aoki et al. 2017; 1029.8 kg m^{-3} , one beaked whale, Aoki unpublished data).

line 492: is there any reference for this statement? I am not sure I believe this given that marine mammals dive to all kinds of depths feeding during all times of year across body sizes successfully.

The references for this statement have been added: “Neutral buoyancy minimises round-trip locomotion cost when diving from/to foraging depth, and locomotion cost during horizontal swimming (Miller et al. 2012, Sato et al. 2013)”.

Neutral buoyancy is considered to minimise locomotion cost (Miller et al. 2012, Sato et al. 2013). Of course, cetaceans make successful foraging dives when they dive and gain energy from feeding. However, they might not need to conduct dives if they have enough energy. The tissue body density of cetaceans is close to neutral buoyancy, or slightly negative. We have added information on the tissue body density of other cetaceans to the manuscript.

line 502: think you mean wintering grounds? As well, if you are going to discuss diving air volume it would be helpful to know how much whales can regulate this during diving as it is something that animals can actually manipulate.

We added references (Yoshida et al. 2016) and modified sentences. Some studies imply several marine animals manipulate air volume (e.g. Yoshida et al. 2016, Sato et al., 2002 J. Exp. Biol. 205, 1189-1197.), but none of study reported air volume dive by dive of cetaceans. We need to develop the accuracy of our methods to estimate air volume dive by dive.

Referee: 2

Comments to the Author(s)

General comments

This is a really nice and robust study, combining two novel approaches to assess body condition in baleen whales. The findings are very convincing and fits previous knowledge of seasonal variation in body condition, differences between reproductive classes etc. I only have some minor comments (see specific comments below). The methods section could benefit from some more details about the UAV measurements, the selection of images and the analyses. Also the result section could provide some more details (see comments below). While the paper is very useful for marine mammal research, perhaps better links to terrestrial mammal could be made in the intro and discussion, to make the paper more interesting to the broader readership of Proceedings of the Royal Society B. Overall though, the study is really nice and will make a very valuable contribution to this field.

We thank the reviewer for providing constructive suggestions. We have added details on the UAV surveys to the Methods and supplements. We have included relevant examples of studies on terrestrial mammals to the Introduction section to draw in the broader readership of Royal Society B. We also emphasize that fatter (larger surface area to length) animals have lower TBD, showing that the “area” gained during the feeding season is mostly made up of lipids and, hence, stored energy, rather than muscle and other heavier tissues.

Specific comments

Lines 29-30: This statement is only true for animals that build up energy reserves. Not all animals do, so please revise.

Based on the reviewer’s comment, we have revised this statement to specifically state marine mammals.

Lines 30-31: Again, not necessarily. This is a very broad statement, which refers to all animals. Be more specific if you are referring to marine mammals.

Based on the reviewer's comment, we have again revised the text to specify marine mammals.

Intro: The intro is good and well structured, but perhaps a bit too focused on marine mammals. The first paragraph talks about "animals" in general, but is very specific to marine mammals, with nearly all references also being marine mammal studies. Could you try to make this one a bit broader, focusing perhaps on mammals in general, but also use some examples from the terrestrial world?

Thank you for these suggestions; accordingly, we have added relevant references of terrestrial mammals and reptiles.

Lines 50-52: Again, this is too generic. Not all animals need to do this. Also there are no references.

As requested, we have incorporated relevant references of other animals.

It would be good to provide a definition of body condition early on in the introduction.

Thank you for this suggestion. We now define body condition the ratio between body lipid and lean body mass. We have added a section explaining how physical body density is used as a measure of body condition.

Lines 52-53: Again, too generic. It seems like you are referring to whales in the abstract and intro, but just write animal. If so, please change to marine mammals.

Based on the reviewer's comments, we have incorporated relevant references on other animals.

Line 60: Change “reserve energy” to “energy reserves”

We rewrote the sentence..

Line 65: Can you provide an estimate for lipid density, or at least a range?

We have added this information based on the reviewer’s request.

Line 66: Same here, can you provide an estimate for lean tissue, or range?

We have added this information based on the reviewer’s request.

Lines 66-68: Not alone. The amount of air in the lungs of the animal should also affect this.

We have added an explanation on residual air. The text now states “Gases in the body also increase positive buoyancy, particularly at shallow depths where they are not compressed”.

Lines 70-71: Change “animals” to “marine mammal” since all the references refers to this.

We changed from “animals” to diving mammals based on the reviewer’s suggestion.

Lines 84-86: You talk about baleen whale migration and reference a seal paper? Please change it.

Thank you for noticing this discrepancy. This has now been addressed.

Lines 88-91: Calf growth rates are also affected by maternal body condition. Check out Christiansen et al. 2018 MEPS.

The information suggested by the reviewer has been added.

Lines 115-116: Add “qualitative” between “Visual” and “assessment”. The word “effective” is a bit odd here. What do you mean with it? Its quick? But is it accurate?

Based on the reviewer’s suggestion, we have added “qualitative”. We have revised the text “Qualitative visual assessments of body condition from photos can provide viable proxies for certain species.”

Line 125: Add latin name for southern right whale: *Eubalaena australis*.

Based on the reviewer’s comment, we have added this information.

Lines 137-140: I would remove these sentences and just embed some of the info, e.g. the distance, into the previous sentence.

Based on the reviewer’s comment, we have shortened the sentence: “Most humpbacks in both locations likely breed in the West Indies during winter”. It is important to provide the information on all sampled humpback whales in this study to demonstrate that they belonged to a single population.

Lines 140-146: I think this sentence deviates from the main aim of your study. I think this can still be discussed in the discussion section, but should not be part of the intro.

As you mentioned, we understand this information deviated from the main aim of this study. But, one of our main results is lower tissue body density of Norwegian whales

than that of Canadian whales during early feeding season. Therefore we would like to inform readers early about differences of migration distance as it is a fundamental difference facing the animals in those two geographical locations.

Material and methods

Line 168: The DJI Phantom 4 was released in 2016 I believe. Does that mean that UAV photogrammetry was not performed during the earlier years of sampling?

As requested, we have added this information: “In 2016-2018, we flew the UAV....”

Lines 172-175: How did the CTD contribute to this study?

We have added the following text to provide this information: “... to measure sea water density near tagged animals....”

Lines 207-208: Please provide a justification for using the compressibility of a northern bottlenose whale for a humpback whale. Is the tissue of a deep diving cetacean comparable to that of a relatively shallow diving humpback?

We now state the compressibility of sea water ($0.447 \times 10^{-9} \text{ Pa}^{-1}$). The humpback whales in our study did not make deep dives often. Consequently, it is difficult to separate the effects of the compressibility of air from the much smaller compressibility of tissue body density. Moreover, the overall compressibility of a mammal's body should be similar to water ($0.46 \times 10^{-9} \text{ Pa}^{-1}$ for water, $0.38 \times 10^{-9} \text{ Pa}^{-1}$ for bottlenose whales). We believe that differences in compressibility between species is very small and is unlikely to influence the estimated values of other parameters.

Lines 240-245: This approach is very similar to Christiansen et al. 2016 Ecosphere, and should be acknowledged. The only difference is that you accounted for body length straight away, whereas Christiansen et al. 2016 did it in a linear model.

As requested, this reference has been added.

Lines 249-251: Why did you only include the surface area from segment 7 (35% BL from rostrum) until 17 (85% BL from rostrum). The latter makes sense, but the start should be at 25%, since that's just behind the eyes, where baleen whales can change quite a lot in width. Alternatively, if you have measurements showing that that is not the case, please provide them to support your cut-off point.

We chose the sections empirically from our measurement of variation across whales at each measurement point using the first set of data we collected (see inserted image). We accept

Photogrammetry method LSSAI

that the full dataset appears to contain more variation in segments 5 and 6 than our initial dataset. However, we believe that including these two section will not make any material change to the results or study conclusions.

In general, the UAV approach needs more details. How did you select your images to be of adequate quality for photogrammetry? The supplementary materials lists the criteria's you used, but not how they were graded and what were deemed to be "good enough" to be included in the analysis. See the supplementary material of

Christiansen et al. (2018 MEPS) for a very detailed description of how UAV images can be graded to assess body condition in baleen whales.

We added the information to both Materials and Methods and supplements : The quality of frames were judged relying on the following predetermined criteria: (1) posture of the individual, (2) brightness of the image and (3) the animal relative to the surface (Table S1). We extracted a few video frames per individual during surfacing periods and allocated scores (1-3, poor to good) of each 3 criteria. The best photo for each whale that had highest averaged score was used to estimate LSSAI.

You also need to provide some measure of error for your UAV measurements. Phantom 4s uses quite a wide angle lens, which is likely to result in quite strong edge effects (distortion). Did you account for this by correcting your image? How large were these measurement errors? Some measurements of a known sized object on land (or sea ideally) could help provide these errors and help determine which images (depending on the position of the animal in the photo) are of good enough quality to be used.

Based on the reviewer's comment, we have added the following information to the supplement: "The DJI Phantom 4 has a wide angled lens that potentially generates strong edge effects (distortion). Since we used relative measures of the projected area versus the length, the absolute value of the metrics did not matter, as long as no significant distortion impacted the relative measurements. To check the extent to which lens distortion potentially impacted our dataset, we measured the length and width (at the pectoral fin) of seven whales when the animal was not positioned in the middle of the image (i.e. the animal was at a corner or on the edge of the frame). We compared the original and corrected images using the distortion coefficient of Phantom 4 (Burnett et al. 2018). For absolute measures of length and width, lens distortion effects were <2%. For the width/length ratio, the distortion effect was <1%. Given that these seven animals had the worst possible positions of all of our data, the estimates of lens-distortion errors were sufficiently small to conclude that lens distortion minimally impacted LSSAI.

”

Results

First paragraph: Could you please provide a bit more details on the animals you sampled, including their age/size class, body length (if known) etc. This is especially interesting for the 20 whales that you both measured with the UAV and tagged. What range of values did this cover, in terms of body condition (from UAV) etc.

We have added a summary of the sex and age classes of the tagged animals, UAV surveyed animals, and the 21 animals for which both measurements were obtained. A table was constructed (Table 1) showing estimates of tissue body density, LSSAI, and the sex/age class of the 21 whales for which both measurements were obtained. Detailed information on all animals is provided in Table S2 and S3 and Fig. 5.

Lines 316-320: Please provide some more details about these findings. What was the effect of these variables on body condition?

We apologise; however, the information requested by the reviewer is unclear. That is the introductory text to a section explaining in the detail the results of a few individuals that we measured on more than one occasion. It is encouraging that those results correspond to what we found in our overall study.

Line 378: You are not really using “width-to-length” ratios, but rather surface area to length. That is an important difference, since your metric is captures more of the animals body shape and size. Don’t sell yourself short□ This is a good metric.

Based on the reviewer’s comment, “width-to-length” has been revised to “projected surface area-to-length ratios”.

Lines 425-426: Provide Latin names for minke, fin and blue whale and for bottlenose dolphin and spotted dolphin.

As requested, this information has been added.

Lines 432-438: I think you can be a bit more explicit here in terms of the implications of your findings for the tissue composition of humpback whales. That fatter (larger surface area to length) animals have lower TBD shows that the “area” gained during the feeding season is mostly made up of lipids, and hence stored energy, rather than muscle and other heavier tissues. This is very interesting, and should be emphasized I think. Perhaps look more into the whaling literature as well and see how the lipid concentration and blubber thickness of different species changes through the feeding season (this should be available for minke, fin and sei whales at least). These are really cool results, so make sure to highlight the importance of them.

Thank you for these suggestions. As suggested, this point has been emphasised in the first section of the discussion.

Lines 451-461: There is a lot of unnecessary detail here. Try to shorten this a bit and get faster to the point you are trying to make (that the migratory distance and consequent timing until returning to the feeding grounds) differs between your two study populations.

We have shorten these sentences. We think it is important to mention other basic information to indicate the differences of tissue body density between Norway and Canada resulted from the differences of migration distance because the differences of tissue body density could be other reasons (e.g. different population and hanging around to search feeding areas).

Line 473: Add “(increased lipid stores)” after “body condition”.

We modified the sentence.

Lines 482-494: It is interesting that the whales seem to end the feeding season in both areas at the same tissue density. While you discuss the benefits of not being too buoyant while feeding, could this also influence migration cost and breeding somehow? Could the locomotion cost during migration be increased in the whales are too buoyant, or could overheating be an issue on the warmer feeding grounds if the whales are too fat when they arrive?

As requested, locomotion cost during migration is now discussed. The text now states “The adjustment of diving air volume, together with negative tissue buoyancy in humpback whales, can yield neutral buoyancy overall at a shallow swimming depth where gases are not highly compressed. Yet more lipid store than we observed would lead to positive buoyancy that gas stores would only make more extreme. Although greater-lipid-store body condition may provide a larger energetic buffer prior to migration, maintaining negative tissue buoyancy might drive the target range of TBD to enable efficient migration.”

Supplementary materials

Line 77: Change “crewless” to “unmanned”.

The requested change has been made.

Lines 110-114: Please provide additional information on how you evaluated this. Did you grade each image (scored it) based on these criterias? If so, what cut-off scores did you use? You mention “angel” to individual? Wasn’t all the photos taken from straight above (with the camera facing down at a 90 degree angle)? If not, that can seriously impact your measurement errors.

We added the information: Video footage was edited using VLC media player 3.0.2. Individual frames were extracted using Free Video to JPG Converter 5.0.101.201. The best quality frames were selected relying on the following predetermined criteria: (1) posture of the individual, (2) brightness of the image and (3) the animal relative to the surface (Table S1). We extracted a few video frames per individuals during surfacing and allocated scores (1-3, poor to good) of each three criteria. The best photo that had highest averaged score was used to estimate LSSAI. Because of rarely flat calm seas and the behavior of the whales on feeding ground, it was more challenging than in their breeding ground areas where they spend substantial time logging at the surface (Christian et al., 2016). Therefore, our measurement might be noisy, as is any empirical measurement, but was sufficient to see variation in individuals (Fig. 2) and the relationship between LSSAI and tissue body density (Fig. 4).

Did your R script account for curved (in the horizontal plane) animals? If not, this could introduce substantial errors in your measurements.

We estimated the projected area of animals. We did not take in account horizontal curvature of animals, as such curvature was not observed.

東京大学 大気海洋研究所

Atmosphere and Ocean Research Institute
The University of Tokyo

www.aori.u-tokyo.ac.jp

Kagari Aoki, Ph.D

Atmosphere and Ocean Research Institute, The University of Tokyo

5-1-5 Kashiwanoha, Kashiwa City, Chiba Prefecture, 277-8564, JAPAN

TEL: +81-4-7136-6226

E-mail: aokikagari@aori.u-tokyo.ac.jp

12 December 2020

I am pleased to resubmit an original article for publication in *Proceedings of the Royal Society B*, titled “**Aerial photogrammetry and tag-derived tissue density reveal patterns of lipid-store body condition of humpback whales on their feeding grounds.**” The paper is coauthored by Kagari Aoki, Saana Isojunno, Charlotte Bellot, Takashi Iwata, Joanna Kershaw, Yu Akiyama, Lucía Martina Martín López, Christian Ramp, Martin Biuw, René Swift, Paul Wensveen, Patrick Pomeroy, Tomoko Narazaki, Ailsa Hall, Katsufumi Sato, and Patrick J. O. Miller.

We have revised the manuscript following as closely as possible the helpful recommendations of the editor and reviewers. We hope that the revised manuscript is now suitable for the publication in *Proceedings of the Royal Society B*.

Yours sincerely,

青木 夏芽里

Kagari Aoki

Reviewer and editor comments are presented in black font, and our responses are presented in green font. Line numbers in a new main document are shown in our comments.

Associate Editor Board Member

Comments to Author:

Your ms has been reviewed by two of the referees are mostly satisfied with your revisions, they have a few more suggestions that should be completed before your paper is accepted. The third referee questioned whether your paper was of sufficient general interest, so I would also recommend revising your ms with that in mind.

Thank you for time and consideration of our manuscript. We have revised the manuscript following as closely as possible the helpful recommendations of the reviewers. We have added another reference to the introduction (Wilder et al., 2016) to highlight the broad interest in studying body condition traits across all taxa. Additionally, we added conclusion text of general interest: “Effective methods for measuring body condition enable evaluation of the fitness consequences of changing environmental conditions and prey availability in the Anthropocene.”

Referee: 3

Comments to the Author(s).

Aoki et al. estimated the body density (a proxy for body condition) of humpback whales based on tag data combined with hydrodynamic modeling. They cross-checked their estimates against independent estimate of relative fatness of the animals examined by drone images. They applied the method to show some regional, seasonal, and inter-individual differences in body condition in this long-migrating species.

The cross validation of two independent methods using a large dataset (>50 individuals) is an important step forward in monitoring the body conditions of large whales in the wild and understanding underlying factors. The regional and seasonal changes in body condition demonstrated in this study are interesting as they have important implications for the migration strategies of this species. These are two main strength of this paper. On the other hand, I have a major concern about the structure of the paper. It has two component (the

method component and ecology component) and the first component is the main one. However, the method for estimating body density based on gliding periods has already been established by a series of studies cited in this paper, including Aoki et al. (2011) that validated the method, although the validation using drone images in large whales is new. For this reason, I believe the main focus of this paper should be shifted to the second, ecology component. Most of the introduction is dedicated to the method component, whereas a large part of the discussion is about the ecology component. This unbalance makes this paper hard to read in the current form.

Thank you for your time and providing useful suggestions. Without neglecting the ecological relevance of our study, we do wish to keep a substantial focus on the methods as our study was primarily designed to validate body condition using two independent methods (UAV and animal-borne tags). Many recent reviews have highlighted the importance of refining and validating body condition indices (Stevenson et al., 2006; Wilder et al., 2016). This evaluation is therefore vital towards investigating body condition in relation to changing environmental conditions and prey availability in the future. Nevertheless, the concurrent analysis of the two types of measurements allowed us to quantify interesting patterns in body condition with location, season and reproductive status, as well as put the measurements indicating negative buoyancy in the context of locomotion costs. We added more emphasis on the ecological interpretation in the last section of the Introduction: We then assess an index of lipid-store body condition derived from these two methods varied in relation to sex, reproductive status, location, and timing within the feeding season.

Specific issues.

Introduction should be condensed, given that the glide method has already been established.

Given the broad readership of the journal, we would prefer to keep the already condensed and concise explanation on marine mammal buoyancy and the glide method. We have however shortened the Introduction by removing discussion on the vertical stratification of baleen whale blubber, and condensed discussion on the Christiansen paper [28].

L286-292. Are the values of drag term and air volume important in this study?

The glide method provides invaluable biological information that would be otherwise challenging to obtain from large, free-ranging cetaceans, and so we feel it is important to mention these to appreciate this methodological advantage.

L336. The opposite argument is possible. The drone method, which is easier, is good enough for estimating animal body conditions?

As the reviewer 3 suggested, we mentioned the argument from drone side: Conversely, the cross-validation confirmed supported the less invasive aerial photogrammetry method, which is widely used in ecology to provide measurements of animals that are difficult to access (L 333-336).

We also note in the Conclusion that one or the other method may be preferred depending on study species and context; “For identifiable resident species, longitudinal measures of body condition are possible using photogrammetry [28]. For poorly individually-marked or less predictable wide-ranging taxa, the tag-based body density method may be most effective for longer-term longitudinal tracking of individuals.”(L480-483)

L452. Many teleosts have neutral buoyancy to remain stationary in the water column, not for efficient locomotion. Highly active fishes, such as tunas, have negative buoyancy.

Good point; we have added “including the cost of remaining at a particular depth”, after “locomotion costs”(L450)

Referee: 2

Comments to the Author(s).

General comments

This is the second time I have the pleasure to review this very nice study, and again I am very supportive of it being published in Proceedings B after just some minor revisions. The study convincingly validates both the glide-method and the photogrammetry method to measure body condition in cetaceans, and will hence be a valuable contribution to this field. The study uses innovative technologies and analytical methods, and a relatively large

sample size (given how difficult it is to obtain paired samples from the same individuals) to support their conclusions. They clearly present their findings in context of earlier work in this area, which hence adds further support to previous work (and vice versa). Minor edits and comments follow below. Overall, a very nice study.

Many thanks again for your time and valuable edits and comments.

Specific comments

Line 31: Add “energy” before “reserves”.

We added it.

Line 33: Provide latin name for humpback whale.

We added it.

Lines 51-52: This statement (“accumulate”) is only true for animals that build up energy reserves. Not all animals do, so please revise.

We modified the sentence: “Accumulating sufficient energy from the environment affects both survival and breeding success for many animal species, and thereby influences the dynamics of entire populations [1]”. We would like to keep the broad perspective to readers as the editor suggested.

Lines 53-54: Again, not necessarily. This is a very broad statement, which refers to all animals. Be more specific if you are referring to marine mammals.

We modified the sentence slightly by inserting “can” before “vary” to be clear this is not always the case, while keeping the perspective of the sentence broad. As before, we would like to keep the perspective of the sentence broad.

Line 57: Add “generally” before “have”, as there are exceptions to this.

We added it.

Line 60: Could you not change “dynamic metric” to “parameter” or “variable” instead. Also

what do you mean with “individual state”, please be more specific.

We changed this to ‘dynamic state variable’ to be specific. Our intention here was to emphasize the role of energy store body condition as a key state variable to understand individual fitness, including its mediating effect on individual behaviour and physiology, and as an integrator of stressor impacts (e.g., from nutritional stress, noise disturbance).

Line 75: Baleen whales also store significant amounts of energy in their muscle, as visceral fats, and also in other tissues (e.g. bones). Whaling data shows that muscle and visceral fats plays a very important role in some species (e.g. minke whales). This needs to be mentioned.

We mentioned this in the paragraph: While most baleen whale lipid reserves are stored in blubber, considerable amounts are also stored in muscle and intra-abdominal fat [4,44]. A unique advantage of the TBD method is that it captures the buoyancy effect of total body lipid-stores simultaneously. (Line 119-121).

Line 97: Add “good” before “predictor”.

We added it.

Line 98: Refs 26 and 27 does not refer to foetal development, but to calf body condition (ref 26) and growth (ref 27). Add Christiansen et al. 2014. Functional Ecology 28: 579-588.

We have very limited space and strict limited numbers of words. Unfortunately, we couldn't make space to add new reference. We deleted “foetal development” as it was not central to our study.

Lines 98-99: Change “body condition, and survival of offspring” to “offspring body condition, growth and survival” to make it clearer that you are referring to the offspring and not adult body condition (since “quantity of energy stored as lipid” is body condition). You can use refs 26 and 27 here.

We corrected it.

Line 125: Add “()” around the latin name to be consistent with earlier formatting. Change “recognize” to “identify”.

We added it.

Lines 130-132: The end of this sentence is strangely worded. Consider revising it to “demonstrating the importance of lipid accumulation during the feeding season for successful breeding”.

We corrected it.

Lines 137-139: Good that you are mentioning this, however it would be even better if you also mentioned it earlier in the intro (see comment to line 75 above).

Thank you very much for your suggestions. We moved this material slightly earlier in the introduction.

Line 142: Add “feeding” before “grounds”.

We corrected it.

Line 145: The distance provided here is confusing. Spell out what it is. Is it the extra distance travelled, the distance of migration one way, or the total round trip distance of the migration.

We corrected it: Consequently, whales that forage in Norway annually migrate 2–3 times further one way trip distance (approximately 8500–9500 km) than whales that forage in Canada.

Lines 146-148: Actually, migration in baleen whales is considered to be relatively cheap (see Braithwaite et al. 2015. Conservation Physiology 3.1: cov001), with the main costs coming from the number of days it takes the whales to migrate (costs of maintenance during those days) rather than from swimming. So the extra costs for the Norwegian whales would come from the additional days it takes them to reach their breeding grounds compared to Canadian whales. Also why would Norwegian whales have lower energy stores at the start

of the feeding season? I would have thought that humpback whales, irrespective of their feeding ground, would aim to time their migration so that they arrive back at the start of the feeding season with a given amount of energy (enough to survive and provide a bit of a buffer). To achieve this while still having a long breeding season, the Norwegian whales would have to feed for longer (or at a faster rate, although it makes little sense why whales wouldn't always be feeding at their fastest rate) and build up larger reserves before departing the feeding grounds, as you propose.

We modified the sentence: Longer duration and migration distance have a greater energetic cost; therefore, we expected that Norwegian whales might have greater lipid stores in late feeding season, but lower stores at early feeding season (L143-146). Our results indicate lower tissue body density of Norwegian whales at early feeding season.

Lines 159-162: Your dates of fieldwork differs a lot between your two study sites, which makes it difficult to test your hypotheses relating to arrival and departure body condition. How did you account for this difference in sampling period?

To clarify, we modified figure 5 and the legend of figure 5: “ Underlying TBD (uTBD) across sex, age classes and reproductive status in Norway and Canada. Each symbol shows individual uTBD estimated from either TBD and/or LSSAI data, coloured by its reproductive status. Both dashed and solid lines indicate decreasing uTBD over feeding seasons predicted by the model (See details for Materials and Methods). The black up arrow shows differences of uTBD between Norway and Canada during the early feeding season. “. We estimated changes in uTBD over Julian date (Fig. 5).

Line 165: Change “the” to “a”.

We corrected it.

Line 239: Write out what sections 7 to 17 represents as % body length from rostrum, e.g. (35-85% body length from the rostrum).

We added the information.

Line 273: Why are you reporting photos of non-tagged animals? Did you use these for your

analyses, or only the photos of the tagged whales? If the latter, then your true sample size was 21 whales.

We used all individuals that include non-tagged animals to estimate uTBD ((c) Underlying tissue body density (uTBD) in relation to date and breeding status in Results).

Line 278: Please change “The global species” to something more appropriate. You only measured two populations of humpback whales in the North Atlantic. Extrapolating this to all humpback whale populations is not appropriate, as there could be morphological and anatomical differences between populations (e.g. the northern hemisphere and the southern hemisphere populations).

Apologies, by global we mean a parameter that is common to all sampled individuals – it essentially describes the individual-average values. This has now been clarified both in the Methods (L 214-217) and Results (L 272).

Lines 295-300: This is confusing. First you report that you obtained repeated measurements from three whales, but then state “these two individuals”. What happened with H607?

ID H607 was also analysed here, but it was previously reported.

To clarify this, We added the information “see Narazaki et al. [32] for adult male ID H607”.

Lines 311-314: This is a very nice finding!

Thank you!

Line 318-322: Very nice!

Thank you!

Line 351: Replace “are” with “have been”.

We made this change as suggested.

Line 376: Provide latin name for fin whale.

We added the latin name.

Lines 397-399: This makes a lot of sense if you consider that humpback whales normally have inter-calving intervals of 2-3 years. Perhaps you could expand on this sentence and speculate on what this means for the inter-calving intervals of humpback whales.

We added this sentence: It might be possible that calving interval would increase if lactating females cannot fully replenish their energy reserves during the year of lactation. Lactation energy expenditure seems to be eventually recovered as our results showed that uTBD of resting females (1037.4 kg m^{-3}) was as low as that of males (1037.2 kg m^{-3}). (L396-L400)

Line 401: Change “animals” to “humpback whales”.

We corrected it.

Lines 426-429: See comment above (Line 145 and Lines 146-148).

We added this information: (one way trip distance, approximately 8500–9500 km)

Lines 431-432: It would be nice to see you expand on this and discuss what it actually means for their migratory behaviour. If whales in Norway are able to return with lower body condition stores and still survive and reproduce, why would Canadian whales not do the same, and stay longer on the breeding grounds. Could there be differences in the feeding rates between the two areas that makes it possible for Norwegian whales to use up more energy during the breeding season compared to Canadian whales.

We hope we can investigate the body condition in relation to migration timing in a future study. Unfortunately, we don't know the exact migration timing from our dataset. I guess that Canadian whales might stay longer on breeding ground. Or, they may come from other feeding area near St. Lawrence or they may go to other feeding area from our study site.

Referee: 4

Comments to the Author(s).

The authors have presented a robust and compelling comparison of two methods for assessing variation in body condition in large cetaceans, a critical metric for understanding prey availability and reproductive success/fitness. I think the authors did a satisfactory job of address the first set of reviewers' comments, in particular with regard to clarifying some UAV methodologies and providing more compelling general context for their study, particularly in the introduction. I only have a few comments to further improve clarity.

Thank you very much. We appreciate your careful and valuable reviews.

Introduction

Line 60-63: Cumbersome run-on sentence, the authors are trying to link several distinct ideas - consider splitting into two or simplifying the wording.

This has now been rephrased” Therefore body condition provides a key dynamic state variable with direct consequences for reproductive output, fitness and demography of free-ranging mammals, with the potential to assess how human activity and environmental changes impact individuals and populations [11,12].”, also addressing comments by Reviewer 2.

Line 75: "energy reserves" instead of "body reserves"

We corrected it.

Line 76-78: List of blubber functions isn't structured properly

We have improved the grammar of the list sentence (Line 86; “Blubber is an important adaptation for aquatic life: it functions as a thermal insulator, contributes to water balance, streamlines the body, and serves as an elastic spring for efficient locomotion”)

Line 91: "a further increase in low-density lipid stores" helps clarify/distinguish this sentence from the previous one.

Thank you for your suggestion. We corrected it.

Line 131-132: move "to successful breeding" to end of sentence.

We corrected it.

Line 133-137: while this information is referred to briefly in the discussion, I do not think it is necessary in the introduction and detracts from the flow of the section. I would remove this entirely and incorporate the remainder of the paragraph (starting with "While most baleen whale...") toward the end of the TBD paragraph (Line 100-117)

We removed that material from the introduction as suggested, also to respond to the comment of reviewer 3.

Results

Lines 266-275: The lengthy lists of number of animals-per-category I find to be an inefficient method for delivering the information. Could all of this information be compiled into a table instead?

We followed the advice of the reviewer and just kept the reference to tables S2 and S3.

Line 421: "whales tracked from Norway have" instead of "Norway has"

Thank you very much for your careful check.

Discussion:

Lines 433-458: in this discussion of cetaceans remaining slightly negatively buoyant (though near neutral) to compensate for lung volume, it may be interesting to contrast them with elephant seals, who DO exhibit positive buoyancy but dive with collapsed lungs and therefore don't have to compensate for buoyant gas. Similarly, is there any evidence of deep-diving cetaceans (sperm or beaked whales) that have lower tissue densities since they also would not have the same air volume to counteract?

The reviewer makes an interesting point, though it is not part of our study. Seals exhale air before dives. And elephant seals often make deep dives more than 100 m, therefore the residual air is collapsed. Sperm whales and beaked whales show slight negative buoyancy.

東京大学 大気海洋研究所

Atmosphere and Ocean Research Institute
The University of Tokyo

www.aori.u-tokyo.ac.jp

More complete information within a cross-species comparison would be needed to reach conclusions, which is beyond the scope of this paper.